# TEMPORAL DIFFERENCE LEARNING:
# WHY IT CAN BE FAST AND HOW IT WILL BE FASTER

**Patrick Schnell, Luca Guastoni and Nils Thuerey**
School of Computation, Information and Technology
Technical University of Munich
Boltzmannstr. 3, 85748 Garching, Germany
{patrick.schnell,luca.guastoni,nils.thuerey}@tum.de

## ABSTRACT

Temporal difference (TD) learning represents a fascinating paradox: It is the prime example of a divergent algorithm that has not vanished after its instability was proven. On the contrary, TD continues to thrive in reinforcement learning (RL), suggesting that it provides significant compensatory benefits. Empirical evidence supports this, as many RL tasks require substantial computational resources, and TD delivers a crucial speed advantage that makes these tasks solvable. However, it is limited to cases where the divergence issues are absent or negligible for unknown reasons. So far, the theoretical foundations behind the speed-up are also unclear. In our work, we address these shortcomings of TD by employing techniques for analyzing iterative schemes developed over the past century. Our analysis reveals that TD possesses a mechanism that enables efficient mapping into the smallest eigenspace—an operation previously thought to necessitate costly matrix inversion. Notably, this effect is independent of the conditioning of the problem, making it particularly well-suited for RL tasks characterized by rapidly increasing condition numbers, e.g. through delayed rewards. Our novel theoretical understanding allows us to develop a scalable algorithm that integrates TD's speed with the reliable convergence of gradient descent (GD). We additionally validate these improvements through a rigorous mathematical proof in two dimensions, as well as experiments on problems where TD and GD falter, providing valuable insights into the future of optimization techniques in artificial intelligence.

## 1 INTRODUCTION

Temporal difference (TD) learning is a training technique for prediction models in multi-step tasks, mostly known for its application to value prediction in reinforcement learning (RL, Sutton, 1988; Kaelbling et al., 1996; Arulkumaran et al., 2017). By estimating expected future rewards, TD learning helps agents make informed decisions based on their interactions with the environment. This approach has proven successful in various domains such as robotics (Littman et al., 1995; Rajeswaran et al., 2017), game playing (Mnih et al., 2015; Lample & Chaplot, 2017), and autonomous driving (Shalev-Shwartz et al., 2016; Sallab et al., 2017), by addressing challenges like delayed rewards where the impact of actions is not immediately clear.

Behind these successes of TD hides a profound contrast in how information is processed over time compared to traditional time prediction methods, like autoregressive models. Such models are 1-step predictors and are often trained via unrolling (Goodfellow et al., 2016), which is an $n$-step update rule. This is suboptimal since predicting $n$ steps with 1-step models leads to exponentially accumulating errors, and an $n$-step update rule requires storing $n$-step trajectories. By contrast, value functions combined with TD offer $n$-step predictors via a 1-step update rule, scoring in terms of both mathematical and computational scalability.

However, challenges arise during optimization: TD objectives are usually minimized with a non-gradient method, harboring the potential for divergence (Baird, 1995). Nonetheless, it often succeeds in finding good solutions quickly. In contrast, provably convergent algorithms such as gradient descent (GD) and its variants are impractically slow in RL, despite being the leading optimization

methods in deep learning. This difference in speed is well-documented empirically (Sutton & Barto, 2018), but the theoretical reasons behind it are unclear. As a consequence, attempts of unifying GD and TD to arrive at a convergent, fast optimization method are often based on intuition. The lack of theoretical understanding means there are no design principles to guide their development.

Our paper addresses this issue with the following contributions:

- We provide a theoretical foundation to explain why TD can be fast starting from the long-established link between condition numbers and the speed of gradient methods. We generalize these ideas to non-gradient methods, such as TD.
- The insights into what makes TD fast uniquely position us to identify the necessary modifications. We demonstrate with a simple method how a unification of GD and TD preserving their positive attributes can look like.

## 2 BACKGROUND

**Optimization Theory**  The natural starting point for all optimization methods based on derivatives is quadratic objectives. They arise in linear systems as well as in nonlinear systems near the optima, where higher-order terms become negligible. Therefore, any method with issues on quadratic objectives will eventually fail. Typically, a quadratic loss $L$ of $n$ variables is expressed as:

$$L = \frac{1}{2}\|Qx\|^2 \quad Q \in \mathbb{R}^{m \times n}, x \in \mathbb{R}^n, \|\cdot\| \ (l_2\text{-norm}) \tag{1}$$

A non-zero target $y \in \mathbb{R}^m$ would not affect convergence properties of iterative solution methods, so we neglect this possibility to maintain compact notation. Such methods take the following generic form in their $t$-th iteration.

$$x_{t+1} = (1 - \eta PQ)x_t \quad \eta \in \mathbb{R} \text{ (learning rate)}, P \in \mathbb{R}^{n \times m} \tag{2}$$

The most prominent examples are GD ($P = Q^T$) and Newton's Method ($P = Q^{-1}$). Convergence occurs if the induced norm of the iteration operator $\|1 - \eta PQ\|$ is strictly smaller than 1. This value is also called the convergence rate because the induced norm, by definition, exactly describes the worst-case decrease of $\|x_{t+1}\|$ relative to $\|x_t\|$, and therefore, the optimization progress. For GD and optimal learning rate, the convergence rate equals $\frac{\kappa-1}{\kappa+1}$, where $\kappa$ is the condition number of the Hessian $Q^TQ$. Ill-conditioned problems ($\kappa \gg 1$) result in a convergence rate only slightly below 1, rendering GD ineffective for solving them (Garrigos & Gower, 2023).

**Ill-Conditioning in Reinforcement Learning**  While one-step tasks can also suffer from ill-conditioning, RL tasks have their own unique mechanisms that elevate condition numbers, thereby complicating optimization, e.g. delayed rewards in multi-step problems require information to flow through several time steps. Suppose we have an $n$-state Markov Reward Process with a linear transition structure ($n \rightarrow n - 1 \rightarrow ... \rightarrow 1 \rightarrow$ terminal), where all rewards are 0 except for the final transition into the terminal state with a reward of 1. This scenario exemplifies a quintessential delayed reward problem, isolated from other complexities in RL such as stochasticity, changing environments, and continuous spaces. The correct values $v$ must fulfill the Bellman equation, a consistency equation stating the value difference between consecutive states equals the intermediate reward. Any violation of this equation is called the temporal difference error $\delta_n$:

$$\delta_n = v(n) - v(n-1) = 0 \text{ for } n > 1 \text{ and } \delta_1 = v(1) - 1 = 0 \tag{3}$$

This linear system is one of the most studied in linear algebra, often called Poisson problem. Its condition number scales as $n^2$ (Strang, 2006). Hence, even though the solution seems trivial—all values are 1—GD becomes increasingly impractical for solving this simple task as $n$ increases.

**Temporal Difference Objective with Function Approximation**  In most RL tasks, the number of states is exponentially large and so is the number of values. Therefore, values $v(s)$ of states $s$ have to be approximated by value functions $v(s, \theta)$ parametrized by $\theta$. As loss function for training serves the TD objective $l$, that is, the squared TD error $\delta$. In the standard RL setting, there is typically a reward $r$ between states $s$ and $s'$, and a discount rate $\gamma$, compared to our example in Equation 3.

$$l = \frac{1}{2}\delta^2 \quad \delta = v(s, \theta) - \gamma v(s', \theta) - r \tag{4}$$

When these terms are summed over all state transitions $(s, s')$, we obtain the full loss for the value function. For linear function approximation, i.e. $v(s, \theta)$ is linear in the parameters $\theta$, this loss is of the form in Equation 1. The $x$ in the notation for the iteration scheme corresponds then to the, possibly shifted, parameters $\theta$, and $Q$ is determined by the state transitions of a specific RL task.

**Gradient Descent in the Temporal Difference Objective**  One way to solve for the TD objective 4 is by applying Gradient Descent, in this context also known as Residual Gradient (Baird, 1995):

$$u_{GD} = -\partial_\theta l = -\delta \cdot \left( \partial_\theta v(s, \theta) - \gamma \partial_\theta v(s', \theta) \right) \tag{5}$$

However, GD is slow as the TD objective is ill-conditioned. This is because on top of the task of finding the correct values, which already has high condition number, comes the secondary task of learning this value through function approximation, making the overall condition number even worse. In Deep Learning, various methods have been proposed to reduce the condition number and so alleviate the slow convergence of GD (Goodfellow et al., 2016). However, they only have limited effectiveness in Deep RL, where typically value functions are approximated: Normalizing data is impractical since the outputs are values and a priori unknown. Also, the input distribution changes continually during the ongoing exploration of the state-action space. As a consequence, initialization schemes, designed to transport normalization properties from one layer to the next, fail since input and output were never normalized. And while momentum can accelerate gradient optimization, it is insufficient to reach an acceptable convergence rate by itself.

**Minimizing the Temporal Difference Objective with Non-Gradient Updates**  An alternative method is the following, commonly referred to as 'Temporal Difference Learning'. From now on, we will use TD to refer to this method. It is important to note that this encompasses not only the temporal difference objective but also the specific form of parameter updates.

$$u_{TD} = -\delta \cdot \partial_\theta v(s, \theta) \tag{6}$$

Compared to $u_{GD}$ in eq. 5, the TD update $u_{TD}$ lacks the derivative term of the subsequent value. Removing this term is motivated by the idea that observations of new rewards should only adjust values of past states. However, the missing term renders the TD update a non-gradient update, therefore lacking any convergence guarantees. That TD updates can indeed lead to divergence has been shown (Baird, 1995). Yet, empirical evidence suggests that this instability sometimes does not occur and that then TD finds solutions in reasonable time Sutton & Barto (2018). From a practical point of view, this ability to deliver solutions in at least some cases makes it the preferred method over consistently slow GD. The logical conclusion is that completely resolving the divergence issue of TD without compromising its speed advantage would be even more beneficial. Achieving this requires a clear understanding of the reasons behind TD's speed advantage.

## 3   THE INCREASED SPEED OF TEMPORAL DIFFERENCE LEARNING

To answer why TD can provide faster convergence than GD, we analyze the iteration scheme introduced in Equation (2), replacing $PQ$ by $H$:

$$x_{t+1} = (1 - \eta H)x_t \tag{7}$$

We begin exactly where previous works stop: The update equation of TD (6) lacks a term present in GD (5), making it a *non-gradient* method. Basic calculus reveals that a gradient field is associated with a corresponding scalar potential, the loss function, whose first derivative is the gradient field and whose second derivative, the Hessian $H_{GD}$, is symmetric. Hence, a non-gradient field like the one from TD need not, and generally will not, have a symmetric Jacobian, allowing for a skew-symmetric part in $H_{TD}$. Gaining insight into how this part affects the eigenvalues is crucial for answering the question of TD's superior convergence speed.

### 3.1   CLOSED-FORM SOLUTION IN TWO DIMENSIONS

To explore the impacts of a non-symmetric $H_{TD}$ on optimization, we first focus on the special case of two-dimensions. This scenario can be solved exactly and will serve as the guiding example

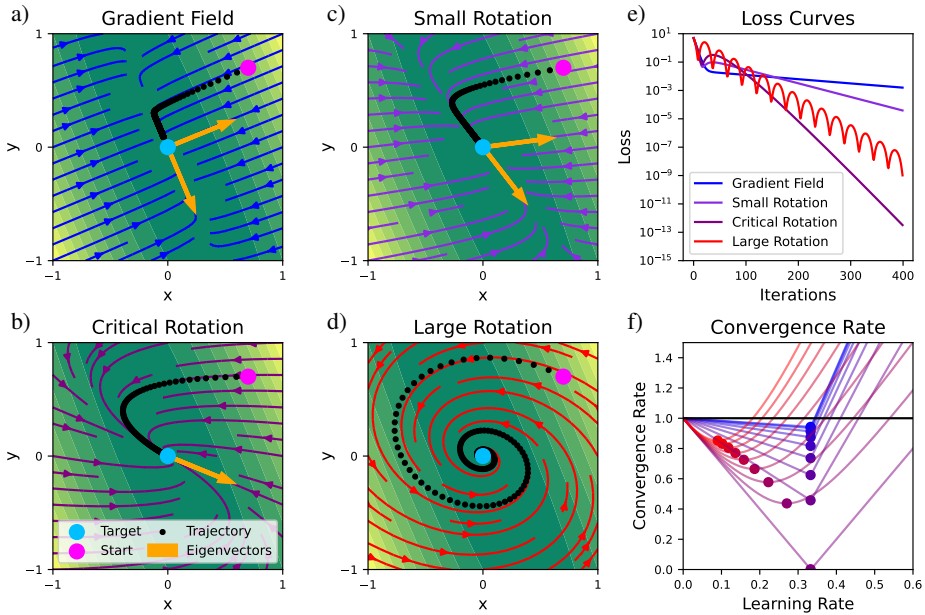

Figure 1: a) - d): Illustration of the four types of a $H_{TD}$ matrix with an example of an optimization trajectory for a random starting point. The background shows the underlying quadratic loss. e) Corresponding loss curves. f) Convergence rate as a function of the learning rate. The amount of rotation increases from blue (no rotation) to red (large rotation). The dot on each curves marks the best convergence rate under optimal learning rate.

for the mathematically precise but abstract argumentation for the generic case. Let us consider the following matrix, which we require to be positive-definite and not the identity for $r = 0$:

$$H = \begin{pmatrix} a & b+r \\ b-r & c \end{pmatrix} \tag{8}$$

This provides a simple model for a possible $H_{GD}$ ($r = 0$) matrix and $H_{TD}$ ($r \neq 0$) matrices with varying strength of the skew-symmetric part. The symmetric part of $H_{TD}$ equals $H_{GD}$, a simplifying assumption for this model that we will remove later. The two eigenvalues $\lambda_{1/2}$ of $1 - \eta H$ are key to understanding the dynamics of the corresponding iteration process.

$$\lambda_{1/2} = 1 - \frac{\eta}{2}(a+c) \pm \frac{\eta}{2}\sqrt{D} \quad \text{with } D = (a-c)^2 + 4b^2 - 4r^2 \tag{9}$$

For $r = 0$, the discriminant $D$ is positive, resulting in two distinct eigenvalues. As $r$ increases in magnitude, $D$ decreases, causing the eigenvalues to converge. With more balanced eigenvalues, the iteration scheme progresses more uniformly in both eigendirections, thereby increasing the convergence rate. We identify four qualitatively distinct cases, illustrated in Figure 1:

a) $r = 0, D > 0$: gradient case, distinct eigenvalues, orthogonal eigenvectors

b) $|r| > 0, D > 0$: small rotation, converging eigenvalues, non-orthogonal eigenvectors

c) $|r| > 0, D = 0$: critical rotation, identical eigenvalues, only one eigenvector

d) $|r| > 0, D < 0$: large rotation, complex eigenvalues with identical real part, complex eigenvectors

Figure 1e depicts the loss curves for the shown optimization trajectories across the four regimes. The $r \neq 0$ curves differ from GD by their short burn-in phase that is followed by convergence at improved speed. In Figure 1f, the convergence rate $c$ is plotted against the learning rate for different values of $r$. Overall, the results indicate that the observed speed-up for $r \neq 0$ is consistent across a wide range of $r$ values, suggesting that this improvement is not a result of specific learning rate choices, but rather stems from the influence of the skew-symmetric part in $H_{TD}$.

## 3.2 GENERALIZING CONVERGENCE RATES

Next, we formalize the ideas above using more rigorous mathematical language. We first link the convergence rate to the minimal eigenvalue, and then show how rotation increases the minimal eigenvalue in arbitrary dimension. We observed that the loss curve may exhibit a burn-in phase or oscillate during its descent, which motivates us to investigate TD's speed through an asymptotic convergence rate $c$, defined as follows:

$$c = \lim_{t \to \infty} \max_{x_0 \in \mathbb{R}^n} \sqrt[t]{\frac{\|x_t\|}{\|x_0\|}} \tag{10}$$

For GD, the connection between $c$ and the condition number arises from the symmetry of $H_{GD}$. For the non-symmetric TD case, establishing a similar link from $c$ to the spectrum of $H_{TD}$ requires further work:

**Theorem 1:** Let $\eta \in \mathbb{R}$, $n \in \mathbb{N}$, $x_0 \in \mathbb{R}^n$, $H \in \mathbb{R}^{n \times n}$ and $t \in \mathbb{N}$. Denote eigenvalues of $H$ and their real and imaginary part as $\lambda_k = R_k + iI_k$ , $k \in \{1, ..., n\}$ and define:

$$\eta_{\max} = 2 \min_k \frac{R_k}{R_k^2 + I_k^2} \qquad R_{\min} = \min_k R_k \tag{11}$$

Then, the iteration scheme $x_t = (1 - \eta H)x_{t-1}$ is convergent if $R_k > 0 \quad \forall k \in \{1, ..., n\}$ and $0 < \eta < \eta_{max}$. Furthermore, the convergence rate $c$ is upper bounded by:

$$c \le \sqrt{1 - \eta R_{\min} \left(1 - \frac{\eta}{\eta_{max}}\right)} \tag{12}$$

**Corollary 1:** Let $R_{\min} \ll 1$, $\rho = 2 \max_k(R_k, I_k)$ and $c_{\text{opt}}$ be the convergence rate under optimal learning rate. Then: $c_{\text{opt}} \le \sqrt{1 - \frac{1}{2}\eta_{\max} R_{\min}} \le 1 - R_{\min}^2/\rho^2 + \mathcal{O}(R_{\min}^4)$

**Corollary 2:** Let $\eta \ll 1$. Then: $c \le 1 - \eta R_{\min} + \mathcal{O}(\eta^2)$

Theorem 1 provides a formula for how the convergence rate can be estimated from the spectrum in the case of a non-symmetric $H_{TD}$. Corollary 1 and 2 offer more interpretable expressions for two relevant situations in deep learning: ill-conditioned matrices ($R_{\min} \ll 1$) and small learning rates ($\eta \ll 1$). Both expressions share the dependence on the eigenvalue with the smallest real part $R_{\min}$, highlighting how increasing this value benefits optimization.

## 3.3 EIGENSPECTRA OF SYMMETRIC, POSITIVE-DEFINITE MATRICES UNDER SKEW-SYMMETRIC PERTURBATIONS

We now generalize the ideas from Section 3.1 to arbitrary $n > 2$. Our analysis begins again with a matrix $H = A + rB$, where $A$ is symmetric, positive-definite and $B$ skew-symmetric. As before, $r = 0$ corresponds to GD and $r \ne 0$ to TD cases. We find the following statements about the spectrum of $H$:

**Theorem 2:** Let $A \in \mathbb{R}^{n \times n}$ be symmetric, positive-definite matrix and $B \in \mathbb{R}^{n \times n}$ skew-symmetric. Additionally, let $A$ and $B$ have non-degenerate eigenvalues. Denote the $i$-th eigenvalue by $\lambda_i(\cdot)$ ordered from smallest to largest eigenvalue, the $i$-th eigenvector of $A$ by $v_i$, and of $B$ by $w_i$. Then:

    i.) $\lambda_{\min}(A) \le \lambda_{\min}^{\text{real}}(A + rB) \le \lambda_{\max}^{\text{real}}(A + rB) \le \lambda_{\max}(A) \quad \forall r \in \mathbb{R}$

    ii.) $\lim_{r \to \infty} \lambda_i^{\text{real}}(A + rB) = \langle w_i, Aw_i \rangle$

    iii.) $\frac{d}{dr}\lambda_{\min}^{\text{real}}(A + rB)|_{r=0} = 0$ and $\frac{d^2}{dr^2}\lambda_{\min}^{\text{real}}(A + rB)|_{r=0} \ge \frac{\langle v_2, Bv_1 \rangle^2}{\lambda_2(A)}$

In Figure 2a, we illustrate these statements for $n = 5$ using random matrices $A$ and $B$. The eigenvalues (blue dotted lines) converge towards each other, as earlier in the $n = 2$ case. Especially the smallest eigenvalue, previously identified as crucial for the convergence rate, improves significantly. The statements of Theorem 2, depicted in red, prove that this picture is representative of

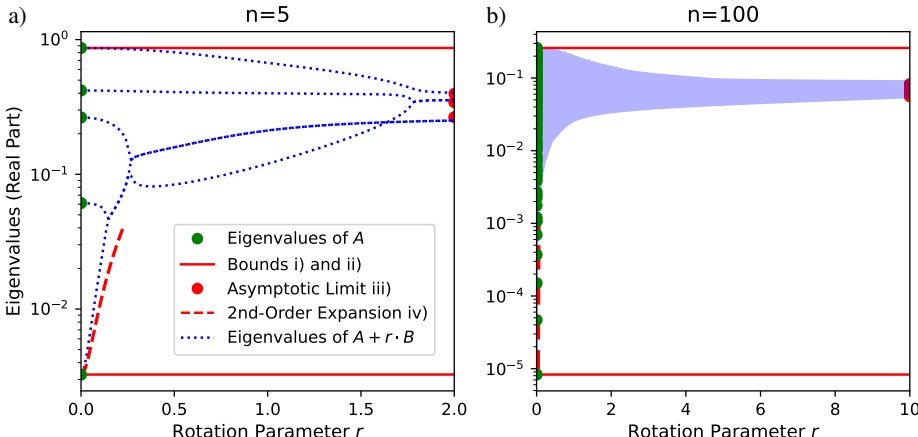

Figure 2: Real-part eigenspectra of a symmetric matrix $A$ with different strengths $r$ of a skew-symmetric perturbation $B$ in a) $n = 5$ and b) $n = 100$ dimensions.

any $A$ and $B$ matrices: Part i) secure that smallest and largest real part eigenvalue cannot separate any further by bounding the spectrum's real part from above and below. Part ii) fixes the right side through a formula for the large $r$ asymptotic limit. Part iii) brings us to the heart of why TD improves convergence speed. The formula quantifies the initial increase of the smallest eigenvector. With the second-smallest eigenvalue serving as the denominator, this term can become extremely large depending on just how ill-conditioned $A$ is. This counters ill-conditioning perfectly, making TD ideally suited for RL. To further illustrate this point, Figure 2b presents a more ill-conditioned example with $n = 100$, showcasing a remarkable rise in the small eigenvalues.

**Enhancing Theorem 2 by Probabilistic Arguments:** Non-degeneracy of eigenvalues is one of the requirements in Theorem 2; however, it is not essential for the statements to hold. Including it simplifies the mathematical proof and, more importantly, allows us to highlight a different type of argument that we will revisit later: In function approximation, the entries of the involved matrices originate from a random process, typically initiated by the random initialization of layers in a neural network or feature matrices in linear function approximation. Among all possible eigenvalues of such a matrix, the occurrence of two identical eigenvalues is a rare edge case and practically irrelevant.

This can be mathematically formalized and proven using measure theory. One of its central theorems states the zero set of a non-constant polynomial is a Lebesgue null set. Building on that, the set of matrices with degenerated eigenvalues is Lebesgue null since these matrices' characteristic polynomials have degenerated zeros if and only if their discriminant, itself a polynomial, is zero. Similarly, part ii) describes the eigenvalues' asymptotic limit which, in theory, could equal $A$'s eigenvalues. But since eigenvectors are defined through polynomial equations, improvement of the spectrum is practically guaranteed. Moreover, by the law of large numbers, as $n$ increases, the eigenvalues of $H$ scatter around the mean of $A$'s eigenvalues, a normal-sized number. In summary, the eigenspectra depicted in Figure 2 are representative of all practically relevant scenarios and this conclusion stands firm under rigorous mathematical evaluation.

### 3.4 THE COMPLETE PICTURE

Finally, we are in a position to present a coherent argument for why TD leads to faster convergence than GD, provided that divergence does not occur.

1. The convergence behavior of iterative methods $x_{t+1} = (1 - \eta H)x_t$ is determined by the matrix $H$, known as the Hessian $H_{GD}$ for GD. For TD, $H_{TD}$ is distinguished by the presence of a skew-symmetric part $B$. We decompose this as $H_{TD} = A + B$.

2. As established in Theorem 1, convergence requires that the real parts of the eigenvalues of $H_{TD}$ are positive. This directly translates to $A$ being positive-definite.

3. With that, we are in the scope of Theorem 2. The real parts of eigenvalues of $H_{TD}$ converge towards each other. Especially the smallest eigenvalue increases quickly, driven by the ill-conditioning of $A$. While we previously thought of $A$ as $H_{GD}$, this assumption is not necessary anymore since Theorem 2 yields a better $H_{TD}$ for any symmetric positive-definite $A$.

4. According to Corollary 1, increasing the smallest eigenvalue improves the convergence rate. However, the speed-up is capped by the maximal learning rate. This limitation becomes critical in the large-rotation regime, where the growing imaginary parts of the spectrum limit the achievable benefits.

5. More importantly, Corollary 2 provides a similar link for small learning rates. This is particularly relevant in nonlinear function approximation, where learning rates must be kept small to ensure the accuracy of linear approximations. In this regime, this explanation for the speed-up from TD is valid without restriction.

This inherent speed advantage of TD has motivated research aimed at addressing its divergence issue. Shortly after Baird introduced GD to RL, he presented interpolation between GD and divergent TD, marking the first attempt to fuse these two methods. Our analysis sheds new light on why this idea does not work: In Theorem 2, we interpolated with a skew-symmetric $B$, whereas in Baird's framework, this matrix would possess an indefinite symmetric part. Perturbation analysis reveals that in this case eigenvalues shift in both directions. Thus, for an ill-conditioned $A$ characterized by small positive eigenvalues, even a minimal negative shift creates negative eigenvalues, making this interpolation method instantaneously divergent. Nevertheless, unifying GD and TD is desirable and we demonstrate now with a simple construction how this can be achieved.

## 4 A PRINCIPLE-GUIDED METHOD TO UNIFY GRADIENT DESCENT AND TEMPORAL DIFFERENCE LEARNING

Inspired by our new understanding, we combine the two update vectors of GD $u_{GD}$ and TD $u_{TD}$ by using GD's sign and TD's magnitude. Therefore, we denote this new method by GDS-TDM:

$$u_{GDS-TDM} = \text{sign}(u_{GD}) \cdot |u_{TD}| \tag{13}$$

The particular form of this update rule is motivated as follows: Through the sign term, the angle between GD and GDS-TDM can be at most $\pi/2$, enforcing movement in similar directions as convergent GD and thereby preventing divergence. Where the signs of GD and TD are constant, the update rule is described by a linear map. Its matrix will have a skew-symmetric part as it is build from the TD update. As we learnt, this is the key property that causes a drastic speed up in ill-conditioned problems. We now present a mathematical proof to show that this theoretical argumentation also holds under mathematical scrutiny:

**Theorem 3 (Enhanced Version):** Let $D \in \mathbb{R}^{2 \times 2}$ be symmetric and positive-definite, $R \in \mathbb{R}^{2 \times 2}$ with eigenvalues unequal 0, $H : \mathbb{R}^2 \to \mathbb{R}^2, x \mapsto -\text{sign}(Dx) \cdot |Rx|$, $S = \{s \in \mathbb{R}^2 | s_i \in \{-1, 0, 1\} \ \forall i \in \{1, 2\}\}$, and for $s \in S$, let $A_s = \{x \in \mathbb{R}^2 | \text{sign}(Dx)_i \cdot \text{sign}(Rx)_i = s_i \ \forall i \in \{1, 2\}\}$. Then:

i.) For all $s \in S$, in $A_s$, $H$ is described by a linear map $R_s \in \mathbb{R}^{2 \times 2}$ such that $\forall x \in A_s : H(x) = R_s x$.

ii.) There exists a maximal learning rate $\eta_{\max} > 0$ such that for $\eta < \eta_{\max}$, the iteration scheme $x_t = (1 + \eta H)(x_{t-1})$ converges to 0.

iii.) If convergence occurs within one $A_s$, the convergence rate $c$ is given by $c = 1 - \eta \lambda_s$, where $\lambda_s$ is the smallest eigenvalue of $-R_s$.

iv.) If convergence occurs across two $A_s$, the convergence rate is given by $c \leq 1 - \eta G(f, K, \epsilon)$, where $f$ is the direction of the smallest eigenvector of $D$, $K$ the condition number of $D$, and $\epsilon$ the angle between $d$ and $Rd$, where $d$ is the boundary vector between the two $A_s$. $G$ is a monotonically increasing function in $K$.

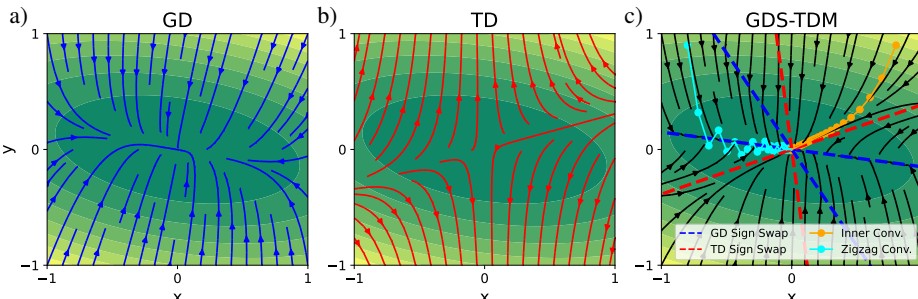

Figure 3: Illustration of our proposed GDS-TDM update: The GD and TD vector fields (left and middle) result in the GDS-TDM field on the right. Convergence either occurs within a single region (orange), or zigzagging (cyan).

As before, we separated the technical requirements that are automatically fulfilled if the involved matrices stem from a continuous sampling process. The full list of these Lebesgue null assumptions can be found in the appendix. Figure 3 gives an illustration of the theorem. Part a) and b) show examples of $D$ and $R$ vector fields, corresponding to GD and TD. Part c) shows the boundaries of the different regions, the dashed lines, along which the sign flips happen. They decompose the domain into the regions $A_s$, each with its own linear map. The orange trajectory shows an example of convergence within one $A_s$, and the cyan trajectory of zigzag convergence between two $A_s$.

To understand why this method can have better convergence speed, consider first the zigzag case (part iv) and the function $G$. For GD, the term after the learning rate decreases with the smallest eigenvalue, explaining its slowness for ill-conditioned $D$. In contrast, here $G$ shows the opposite behavior, increasing convergence speed with the condition number of $D$. As one can also guess from Figure 3c, the dependence depends more on geometric quantities of $R$, not $D$, such as the angle at which the flow lines of $R$ intersect the boundary line. For the inner convergence case (part iii), the convergence rate depends on the smallest eigenvalue. The linear maps $R_s$ are non-symmetric, and as we have shown before, this substantially increases the smallest eigenvalue, improving convergence.

Besides this mathematical characterization, practical advantages of the update rule are that it is easy to implement and causes minimal computational overhead. Since GD and TD are two backpropagation techniques based on the same forward pass, they can be computed in parallel with little additional memory. The runtime increase due to a component-wise sign comparison is likewise minimal.

## 5 EXPERIMENTS

While our central contribution is a theoretical foundation for understanding TD and for deriving algorithms with well-understood properties, we provide a first set of empirical results for how GDS-TDM compares against GD and TD. In our experiments, we target value estimation, which is a fundamental part of most practical reinforcement learning algorithms. The choice and setup of these experiments are described below and motivated by our intention to provide a clear consistency check of our theoretical derivation. Two additional experiments can be found in Appendices B and C.

**Two-State Example for TD's Divergence**  We consider a variant of Tsitsiklis & Van Roy (1996b)'s canonical value prediction task with a second parameter, as shown in Figure 4a. There are two transitions (state $0 \to 0$ and $1 \to 0$). The self-transition is convergent while the other one leads to divergence. Training off-policy using a state distribution with too much weight on the divergent transition will therefore make TD divergent on this task. This alternating between convergent and divergent updates can be nicely seen in TD trajectory in Figure 4b. GD and GDS-TDM both converge. However, as can be seen in Figure 4c, the convergence rate of GDS-TDM is substantially better, agreeing with our theoretical treatment.

**10x10 Grid World**  This is a classic RL environment, for which value functions are easy to visualize, allowing for easier assessment of iterative solution methods. We consider a two-dimensional non-periodic grid with terminal states located in the top-left and bottom-right corners. The agent

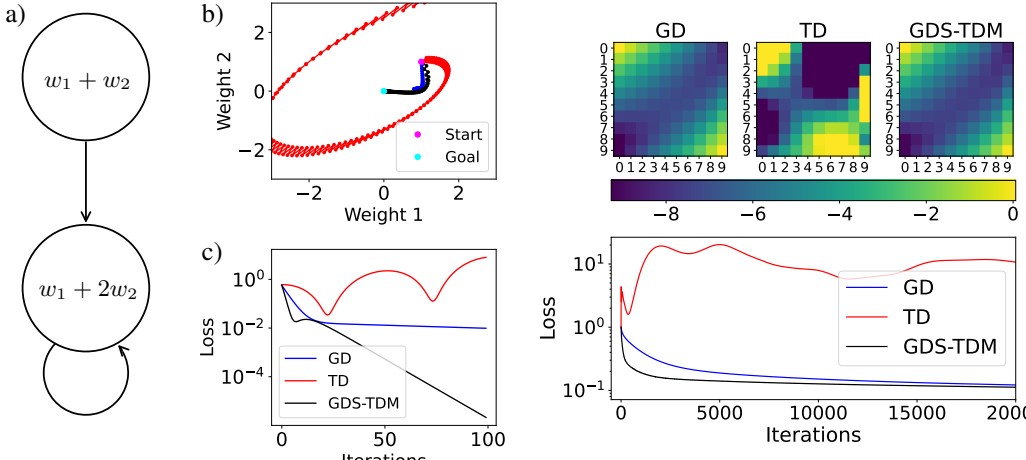

Figure 4: a) Schematic representation of Tsitsiklis and Van Roy's counterexample. b) Learning trajectories of the different algorithms in the parameter space. c) Training loss as a function of the learning iterations.

Figure 5: (Top) Visualization of the learnt value function in a $10 \times 10$ grid world for the different methods: gradient descent (GD, left), temporal difference (TD, center) and GDS-TDM (right). (Bottom) Comparison of the learning curves of the three methods.

can move up, down, left and right (unless on the boundary), and is penalized with a reward of $-1$ for every move that does not lead to the terminal state. Given the optimal policy, we estimate its values using a 100-parameter linear function approximator based on polynomial features, which we train on full batches to avoid stochastic effects of mini-batch training. The state pairs for the updates are sampled with equal probability, i.e. this is an off-policy case. In the optimal policy distribution, state pairs closer to the terminal states would occur more frequently than those further away. The exact solution has a value of $0$ in the terminal corners and is minimal along the diagonal. An illustration is given in Figure 5, where GD and GDS-TDM correctly approach this solution. In contrast, TD suffers from divergence and is unable to find a good answer. Comparing the loss curves, we observe that GD and GDS-TDM behave similarly, both steadily decreasing the loss and eventually reaching a similar accuracy. This 100-parameter example gives us an outlook beyond the strict mathematical argumentation of our 2D proof. The absence of a substantial speed-up of GDS-TDM over GD could point to a limitation of our method in higher dimensions, suggesting that additional modifications might be necessary. It is also possible that Grid World, with its immediate rewards, does not generate the high condition numbers where TD methods thrive and GD fails. The fact that the value function estimated by GD already closely resembles the exact solution supports this. Regarding divergence, the picture is clearer: GDS-TDM does not encounter the divergence issues exhibited by TD.

## 6 RELATED WORK

A number of successful applications of TD exists in literature (Tesauro, 1995; Mnih et al., 2015). TD learning was originally introduced in contrast to Monte-Carlo methods, which rely on complete episodes, along with further TD variants that vary the number of steps over which errors are evaluated (Sutton, 1988). Notably, our analysis applies to all variants of TD, as we did not use any specific form of updates but only their non-gradient nature. The fact that TD is not a gradient method and divergent was discovered by Baird (1995), and another example of divergence of TD was given by Tsitsiklis & Van Roy (1996b). While the precise circumstances under which TD diverges remain mostly unclear, it has been shown that TD converges for linear function approximation with on-policy training Tsitsiklis & Van Roy (1996a). In that case, the convergent transitions outweigh the divergent ones. Generalizations of linear TD's on-policy convergence were found by Asadi et al. (2023). Even in this setting, a method convergent on every transition could speed-up optimization. Emphatic-TD methods (Sutton et al., 2016) reweight the state distribution to update less frequently on divergent state transitions. While this can suppress divergence, it inherently impairs the ability

to learn the divergent part of the state space. The factors contributing to divergence in RL were later summarized under the term 'deadly triad', including TD training, function approximation and off-policy training. Given the importance of these elements in modern RL, a number of different research works have tried to address the shortcomings of TD divergence. For instance, van Hasselt et al. (2018) try to build intuition for the TD algorithm behaviour and design a number of mitigation strategies that they test empirically. Furthermore, least-squares TD (LSTD), in its recursive (Bradtke & Barto, 1996) or incremental (Geramifard et al., 2006) implementation, provide convergence guarantees at the cost of a higher computational complexity. More recently, diverse attempts are made to address TD's divergence through normalization techniques (Gallici et al., 2024), or by introducing and updating a second set of parameters that then leads to updates on the actual network parameters (Wang & Ueda, 2022). Beyond the divergence issue, Kumar et al. (2021) studies how regularization can improve the quality of TD solutions.

The introduction of GD to RL, under the name residual gradient (Baird, 1999), has been proposed as a solution to the divergence of TD. This was quickly followed by interpolation techniques between TD and GD to address the apparent slowness of GD (Baird, 1999), marking the first attempt to combine the positive elements of both methods into a new method. Ill-conditioning, the property that slows down GD, was also investigated in an RL context, confirming that condition numbers can indeed be high in RL tasks (Sharifnassab & Sutton, 2023). Further attempts to unify TD and GD have been made under the name gradient TD (GTD) methods (Sutton et al., 2008; 2009). They address another phenomenon in how TD and GD can differ: On stochastic tasks, GD can converge to a different solution than TD. While GTD methods converge and approach the original TD solution, they do not address the slowness of GD. This is apparent when considering that the slowness issue already exists in deterministic tasks, where GTD simplifies to GD. Further improvements to the GTD algorithms were proposed by Yao (2023) and Qian & Zhang (2023), reducing the number of hyper-parameters to tune in the algorithm.

It is noteworthy that positive observations of learning with non-gradient methods reach beyond RL (Schnell & Thuerey, 2024). Examples include time series prediction with unrolled computation graphs (Stachenfeld et al., 2022) often in combination with differentiable simulators (Um et al., 2020), as well as in the context of bilevel optimization (Bolte et al., 2024; Domke, 2012), such as hyperparameter optimization (Lorraine et al., 2020), deep equilibrium models (Geng et al., 2021; Fung et al., 2022) and meta-learning (Andrychowicz et al., 2016). Typically, the backpropagation pass is either shortened or modified in another way, which consequently destroys the gradient property of the outcome, thus falling within the scope of our argument. Similar to TD, the evidence is primarily empirical and these research fields could benefit also from the discovery of the underlying reasons behind the observed improvements.

## 7 CONCLUSION

Our work presented a theoretical framework that explains the widespread and puzzling observations that TD can deliver a crucial speed-up for making RL tasks solvable. We identified the skew-symmetric part of the iteration operator, which distinguishes non-gradient from gradient-based methods, as the key quantity driving this acceleration. An interesting future direction will be to design model architectures that effectively control this quantity to enable a consistent performance boost. This shares similarities with initialization schemes that are tailored to influence the condition number to enhance gradient descent.

Furthermore, we proposed a method that combines the convergence properties of GD with the speed-up mechanisms of TD. We supported this with a mathematical proof for the two-dimensional case, providing a detailed understanding of how these properties integrate. With our focus primarily on the theoretical aspects, a natural next step is further empirical investigations. These could include exploring nonlinear function approximation, testing compatibility with momentum and other optimization techniques, and extending to advanced control algorithms in RL.

In an empirically-driven field, it is crucial to remember the significance of theoretical results. Unifying GD and TD has been a longstanding goal in RL, and despite extensive empirical research, it has not been satisfactorily resolved, even in two dimensions. In that spirit, we believe our theoretical framework can serve as a stepping stone for developing more reliable and faster RL algorithms.

ACKOWLEDGEMENTS

This work was supported by the ERC Consolidator Grant *SpaTe* (CoG-2019-863850). We would also like to express our gratitude to the reviewers and the area chair for their helpful feedback.

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

# A MATHEMATICAL PROOFS

## A.1 THEOREM 1 AND ITS COROLLARIES

**Theorem 1:** Let $\eta \in \mathbb{R}$, $n \in \mathbb{N}$, $x_0 \in \mathbb{R}^n$, $H \in \mathbb{R}^{n \times n}$ and $t \in \mathbb{N}$. Denote eigenvalues of $H$ and their real and imaginary part as $\lambda_k = R_k + iI_k$, $k \in \{1, ..., n\}$ and define:

$$\eta_{\max} = 2 \min_k \frac{R_k}{R_k^2 + I_k^2} \qquad R_{\min} = \min_k R_k \tag{14}$$

Then, the iteration scheme $x_t = (1 - \eta H)x_{t-1}$ is convergent if $R_k > 0 \quad \forall k \in \{1, ..., n\}$ and $0 < \eta < \eta_{max}$. Furthermore, the convergence rate $c$ is upper bounded by:

$$c \leq \sqrt{1 - \eta R_{\min}\left(1 - \frac{\eta}{\eta_{max}}\right)} \tag{15}$$

**Proof:**

We start with:

$$
\begin{aligned}
c &= \lim_{k \to \infty} \max_{x_0 \in \mathbb{R}^n} \sqrt[k]{\frac{\|x_k\|}{\|x_0\|}} \\
&\leq \lim_{k \to \infty} \sqrt[k]{\|(1 - \eta H)^k\|} \\
&= \max_{k \in \{1, ..., n\}} |1 - \eta \lambda_k|
\end{aligned}
\tag{16}
$$

The first equality is the definition of the convergence rate; the following inequality rewrites the expression using operator norms; the last equality applies Gelfand's formula (Lax, 2002). For the inputs to max-operator, we have:

$$|1 - \eta \lambda_k| = \sqrt{1 - 2\eta R_k + \eta^2\left(R_k^2 + I_k^2\right)} \tag{17}$$

Hence, in order to have $c < 1$, we require $R_k > 0$ for all $k \in \{1, ..., n\}$. The radicand is a parabola with a value of 1 for $\eta = 0$. Hence, it is also 1 for $\eta = 2\eta_k^c$, where $\eta_k^c$ is the minimum.

$$\frac{d}{d\eta}\left(1 - 2\eta_k^c R_k + \eta_k^c\left(R_k^2 + I_k^2\right)\right) = 0 \tag{18}$$

We find:

$$n_k^c = \frac{R_k}{R_k^2 + I_k^2} \tag{19}$$

Hence, $c < 1$ for $0 < c < \eta_{\max} = 2 \min_k n_k^c$.

For the inequality on the convergence rate, we use again properties of parabolas: parabolas are determined by three conditions. A parabola $P(\eta)$ lying above the $k$ other parabolas between 0 and $\eta_{\max}$ is fixed by:

- $P(0) = 1$
- $P(\eta_{\max}) = 1$
- $\frac{d}{d\eta}P(0) = -2R_{\min}$

This leads to:

$$P(\eta) = \frac{2R_{\min}}{\eta_{\max}}\eta^2 - 2R_{\min}\eta + 1 \tag{20}$$

**Corollaries 1 and 2:**

Both follow from Taylor expansion and determining the extrema of Equation 20.

## A.2 THEOREM 2

**Theorem 2:** Let $A \in \mathbb{R}^{n \times n}$ be symmetric, positive-definite matrix and $B \in \mathbb{R}^{n \times n}$ skew-symmetric.. Additionally, let $A$ and $B$ have non-degenerate eigenvalues. Denote the $i$-th eigenvalue by $\lambda_i(\cdot)$ ordered from smallest to largest eigenvalue, the $i$-th eigenvector of $A$ by $v_i$, and of $B$ by $w_i$. Then:

i.) $\lambda_{\min}(A) \leq \lambda_{\min}^{\text{real}}(A + rB) \leq \lambda_{\max}^{\text{real}}(A + rB) \leq \lambda_{\max}(A) \quad \forall r \in \mathbb{R}$

ii.) $\lim_{r \to \infty} \lambda_i^{\text{real}}(A + rB) = \langle w_i, A w_i \rangle$

iii.) $\frac{d}{dr} \lambda_{\min}^{\text{real}}(A + rB)|_{r=0} = 0$ and $\frac{d^2}{dr^2} \lambda_{\min}^{\text{real}}(A + rB)|_{r=0} \geq \frac{\langle v_2, B v_1 \rangle^2}{\lambda_2(A)}$

**Proof:**

**Part i)**

We compute the numerical range of $A + rB$, which is defined as the range of the Rayleigh coefficient: Let $x \in \mathbb{C}^n$ and normalized. Denote the vectors of the eigenbasis of $A$ by $v_j$ and the corresponding eigenvalues by $\lambda_j$.

$$
\begin{aligned}
\langle x, (A + rB)x \rangle &= \sum_j \lambda_j |x_j|^2 + r \cdot \sum_{j,k} B_{jk} \bar{x}_j x_k \\
&= \sum_j \lambda_j |x_j|^2 + r \cdot \sum_{j,k} B_{jk} (x_j^R x_k^R + x_j^I x_k^I) + ir \cdot \sum_{j,k} B_{jk} (x_j^R x_k^I - x_j^I x_k^R)
\end{aligned}
\tag{21}
$$

Since $B$ is skew-symmetric, the second term adds to zero. Therefore, an expression for the real part of the Rayleigh quotient contains only the first term. Using $\|x\|_2 = 1$, we find a bound for the numerical range:

$$
\lambda_{\min}(A) \leq \text{Re}(\langle x, (A + rB)x \rangle) \leq \lambda_{\max}(A)
\tag{22}
$$

As the numerical range of a matrix contains its spectrum, the claim follows.

**Part ii) and iii):**

We follow the procedure outlined by Lax (2013) to compute the derivatives: Consider a differentiable square-matrix-valued function $F(t)$ of a real variable $t$. Let $\mu$ be a non-degenerate eigenvalue of $F(0)$. Then for sufficiently small $t$, $A(t)$ has an eigenvalue $\mu(t)$ and corresponding eigenvector $h(t)$ that both depend differentiably on $t$. The derivatives with respect to $t$ at $0$ are denoted by dots and given as follows with $h = h(0)$ and $l$ the left eigenvector of $F(0)$ corresponding to $\mu$:

$$
\dot{\mu} = \frac{\langle l, \dot{F} h \rangle}{\langle l, h \rangle}
\tag{23}
$$

$$
(F(0) - \mu) \dot{h} = -(\dot{F} - \dot{\mu}) h
\tag{24}
$$

$$
\ddot{\mu} = \frac{\langle l, \ddot{F} h \rangle + 2 \langle l, \dot{F} \dot{h} \rangle + 2 \dot{\mu} \langle l, \dot{h} \rangle}{\langle l, h \rangle}
\tag{25}
$$

**Application to iii):**

$A + rB$ is a differentiable square-matrix-valued of $r$ equalling $A$ for $r = 0$. $A$ is symmetric, implying left and right eigenvectors are identical. Using $A v_j = \lambda_j v_j$ to denote eigenvalues and normalized eigenvectors, we find:

$$
\dot{\lambda}_j = \langle v_j, B v_j \rangle = 0
\tag{26}
$$

The last equality follows from the antisymmetry of $B$.

$$(A - \lambda_j)\dot{v}_j = -(B - \dot{\lambda}_j)v_j \tag{27}$$

This singular system is solved by the following expression for any $\alpha \in \mathbb{R}$. Note that the inverse here is the pseudoinverse.

$$\dot{v}_j = -(A - \lambda_j)^{-1}(B - \dot{\lambda}_j)v_j + \alpha v_j \tag{28}$$

The second derivatives are given as follows:

$$
\begin{aligned}
\ddot{\lambda}_j &= 2\langle v_j, B\dot{v}_j \rangle \\
&= 2\langle Bv_j, (A - \lambda_j)^{-1}Bv_j + \alpha v_j \rangle \\
&= 2\langle Bv_j, (A - \lambda_j)^{-1}Bv_j \rangle \\
&= 2\sum_{k \neq j} \frac{\langle v_k, Bv_j \rangle^2}{\lambda_k - \lambda_j}
\end{aligned}
\tag{29}
$$

For the largest eigenvalue, all denominators are negative and therefore the largest eigenvalue does not grow for sufficiently small $r$. For the smallest eigenvalue, the situation is reversed.

$$
\begin{aligned}
\ddot{\lambda}_{\max} &\leq 0 \\
\ddot{\lambda}_{\min} &\geq 0
\end{aligned}
\tag{30}
$$

**Application to ii):**

In the same spirit, we apply the formulas for the derivatives again. $B$ is skew-symmetric, therefore diagonalizable. We have:

$$\frac{d}{d\epsilon}\lambda_i(\epsilon A + B) = \epsilon\langle w_i, Aw_i \rangle \tag{31}$$

Since eigenvalues of skew-symmetric matrices lie on the imaginary axis, we find:

$$\lambda_i(\epsilon A + B) = \epsilon\langle w_i, Aw_i \rangle + \mathcal{O}(\epsilon^2) \tag{32}$$

Hence, we can verify the claim:

$$\lim_{r \to \infty} \lambda_i(A + rB) = \lim_{\epsilon \to 0} \frac{1}{\epsilon}\lambda_i(\epsilon A + B) = \langle w_i, Aw_i \rangle \tag{33}$$

### A.3 THEOREM 3

**Theorem 3:** Let $D \in \mathbb{R}^{2 \times 2}$ be symmetric and positive-definite, $R \in \mathbb{R}^{2 \times 2}$, $H : \mathbb{R}^2 \to \mathbb{R}^2, x \mapsto -\operatorname{sign}(Dx) \cdot |Rx|$, $S = \{s \in \mathbb{R}^2 | s_i \in \{-1, 0, 1\}\}$, and for $s \in S$, let $A_s = \{x \in \mathbb{R}^2 | \operatorname{sign}(Dx)_i \cdot \operatorname{sign}(Rx)_i = s_i \ \forall i \in \{1, 2\}\}$. Then:

    i.) For all $s \in S$, in $A_s$, $H$ is described by a linear map $R_s \in \mathbb{R}^{2 \times 2}$ such that $\forall x \in A_s :$ $H(x) = R_s x$.

Assume further that $D$ has different eigenvalues, $R$ has eigenvalues unequal $0$, and that all the $R_s$ are diagonalizable. Let $v_{1,s}$ be the maximal eigenvector to $R_s$ (eigenvector to the eigenvalue with largest real part). We assume $v_{1,s}$ and $Dv_{1,s}$ have no zero components for all $s$. Denote by $d_i$ a vector for which the $i$-th component of $Dd_i$ is $0$. Then we assume $Rd_i$ has no zero component.

ii.) There exists a maximal learning rate $\eta_{\max} > 0$ such that for $\eta < \eta_{\max}$, the iteration scheme $x_t = (1 + \eta H)(x_{t-1})$ converges to 0.

iii.) If convergence occurs within one $A_s$, the convergence rate $c$ is given by $c = 1 - \eta \lambda_s$, where $\lambda_s$ is the smallest eigenvalue of $-R_s$.

iv.) If convergence occurs across two $A_s$, the convergence rate is given by $c \leq 1 - \eta G(f, K, \epsilon)$, where $f$ is the direction of the smallest eigenvector of $D$, $K$ the condition number of $D$, and $\epsilon$ the angle between $d$ and $Rd$, where $d$ is the boundary vector between the two $A_s$. $G$ is a monotonically increasing function in $K$.

**Proof:**

**Description by a Linear Map**   For all $s \in S$, the matrix elements of $R_s$ are given by $(R_s)_{ij} = -s_i \cdot R_{ij}$. Then, for all $k \in \{1, 2\}$:

$$(R_s x)_k = \sum_{j=1}^{2} -s_k R_{kj} x_j = -s_k \cdot (Rx)_k \tag{34}$$

Furthermore, for all $x \in A_s$:

$$H(x)_k = -\operatorname{sign}(Dx)_k \cdot |Rx|_k = -s_k \frac{|Rx|_k}{\operatorname{sign}(Rx)_k} = -s_k \cdot (Rx)_k \tag{35}$$

Therefore, for all $x \in A_s$, $H(x) = R_s x$.

**Further Subdividing $\mathbb{R}^2$**   For each $x \in \mathbb{R}^2$, there exists an $s$ such that $x \in A_s$. We will classify the dynamics within each region $A_s$ by using the eigendecomposition of its linear map $R_s$:

$$R_s v_k = \lambda_k v_k \tag{36}$$

We choose the numbering according to the size of the real part of eigenvalues:

$$\Re(\lambda_1) \geq \Re(\lambda_2) \tag{37}$$

The division $A_s$ of $\mathbb{R}^2$ consists of straight lines (when a component of $s$ is zero) and double cones (for the remaining $s$). A cone $C$ generated by vectors $g_i$ is defined as:

$$C = \{x \in \mathbb{R}^n | x = \sum_i \alpha_i g_i \text{ where all } \alpha_i > 0\} \tag{38}$$

A double cone is the unification of the two cones generated by $g_i$ and $-g_i$. The problem with this division is that the behavior within one specific cone can vary. For instance, an $A_s$ with only negative eigenvalues and the eigenvectors inside of $A_s$, there will have some $x$ for which it converges to 0 and other $x$ for which it leaves $A_s$. This is why we further subdivide the $A_s$ into subregions $B$ that behave consistently for all points inside. This works as follows:

1. First we split all double cones into the two cones they are made of.

2. For each cone $C$, we check if an eigenvector $v$ of the corresponding $R_s$ is in the interior of $C$. If not, we leave $C$ as it is. If so, we divide $C$, generated by $\{g_1, g_2\}$ into new cones always replacing one of the generators by $v$: $\{v, g_2\}$ and $\{v, g_2\}$. In between these new smaller cones, we will also have 1-dimensional cones, generated by fewer than $n$ vectors.

3. If there still are eigenvectors in the two smaller cones, we repeat this procedure until there are none inside anymore.

This will leave us with a subdivision of $\mathbb{R}^n$ into subregions $B$ with no eigenvectors in their interiors.

**Restriction to Neighbor Transitions**   For our convergence proof to work, we restrict our iteration scheme to avoid jumping between subregions but only transition to adjacent subregions. Adjacent subregions $B_1$ and $B_2$ means $\partial B_1 \cap \partial B_2 \neq \{0\}$. For a subregion $B$, we achieve this by allowing only learning rates smaller than a maximal learning rate $\eta_B$. Let $B$ be generated by $g_i$ and $C$ be $B$ unified with all its adjacent subregions. Then we define $D_\delta$ as the cone spanned by the generators $g_1 - \delta \cdot g_2$ and $g_2 - \delta \cdot g_1$. These are obviously cones containing $B \subset D_\delta$ for all $\delta \geq 0$.

Now, there exists a $\gamma > 0$ s.t. $D_\gamma \subset C$. For each of the $g_i$, determine $\eta_i$ such that $f(\eta) = (1 - \eta R_B)g_i \in D_\gamma$. This $\eta_i$ is greater than 0 since $f$ is continuous, $f(0) = g_i$ and $g_i$ is an interior point of $D_\gamma$. Set $\eta_B = \min_i \eta_i$. Then for any $x \in B$, $x = \alpha_1 g_1 + \alpha_2 g_2$, the result of an iteration step $y$ is:

$$(1 - \eta_B R_B)x = \sum_{i=1}^{n} \alpha_i(1 - \eta_B R_B)g_i \tag{39}$$

Since $\alpha_i > 0$ due to $x \in B$, $y$ is in the open cone spanned by $(1 - \eta_B R_B)a_i$. These generators are in $D_\gamma$ by construction. Hence, $y \in D_\gamma$ and $y \in C$, This means hat for all learning rates below $\eta_B$, the iteration scheme maps only to $B$ or its adjacent subregions. If the subregion was a lower-dimensional cone, we can apply this argument to an adjacent full-dimensional cone. Choosing the minimum $\eta_B$ over all subregions, we receive a maximal learning rate $\eta_{\text{max, AT}}$ that guarantees adjacent transitions globally.

**Classification of the Dynamics within Subregions**  We classify a subregion $B$ according to the position of the largest eigenvector $v_1$ of their corresponding linear map $R_B$.

**Case 1:** $v_1 \in \partial B$ **, convergence**

($v_1$ lies on the boundary of $B$)

An immediate consequence is that the eigenvalue $\lambda_1$ to $v_1$ has to be negative ($\lambda_1 = 0$ is not possible by our theorem assumptions). To prove this, let $\{x_t\}$ be sequence of vectors converging to $v_1 = \lim_{t \to \infty} x_t$ with $x_0 \in B \quad \forall k \in \mathbb{N}$. Then by how our method is constructed:

$$\text{sign}(Dx_t) = -\text{sign}(R_B x_t) \quad \forall t \in \mathbb{N} \tag{40}$$

By continuity, we have $\lim_{t \to \infty} R_B x_t = R_B v_1 = \lambda_1 v_1$ and $\lim_{t \to \infty} Dx_t = Dv_1$. Since by assumption of the theorem $v_1$ and $Dv_1$ have no zero components, and the sign function is continuous outside of 0, we also have $\lim_{t \to \infty} \text{sign}(R_B x_t) = \text{sign}(R_B v_1) = \text{sign}(\lambda_1 v_1)$ and $\lim_{t \to \infty} \text{sign}(Dx_t) = \text{sign}(Dv_1)$. Putting that together, we find:

$$\text{sign}(Dv_1) = -\text{sign}(R_B v_1) = -\text{sign}(\lambda_1)\text{sign}(v_1) \tag{41}$$

By assumption of our theorem, $D$ is positive definite, implying the angle between $v_1$ and $Dv_1$ has to be smaller than $\pi/2$ and eliminating the possibility of a positive $\lambda_1$ in the last equation. Therefore, $\lambda_1$ is negative.

Next, we show that our method converges to 0 from within this region: For all $x \in B$, we can write $x = \sum_{i=2}^{n} \alpha_i g_i$. This is again the standard parametrization of cones with $\alpha^i$ positive and $g_i$ the generators of the cone. Note that $v_1$ is such a generator; we set $g_1 = v_1$. Furthermore, this parametrization also includes lower-dimensional cones, where the $g_i$ could be linear dependent. Let now be $x_0 \in B$. We obtain the next iteration vector $x_1$ by applying the iteration scheme with the linear operator of $B$:

$$\begin{aligned}
x_1 &= (1 + \eta H)(x_0) \\
&= (1 + \eta R_B)x_0 \\
&= (1 + \eta R_B)\sum_{i=1}^{2} \alpha_i g_i \\
&= (1 + \eta R_B)(\alpha_1 v_1 + \alpha_2 \beta_1 v_1 + \alpha_2 \beta_2 v_2) \\
&= (1 + \eta R_B)((\alpha_1 + \alpha_2 \beta_1)v_1 + \alpha_2 \beta_2 v_2) \\
&= (1 + \eta\lambda_1)(\alpha_1 + \alpha_2\beta_1)v_1 + (1 + \eta\lambda_2)\alpha_2\beta_2 v_2 \\
&= (1 + \eta\lambda_1)(\alpha_1 + \alpha_2\beta_1)g_1 + (1 + \eta\lambda_2)\alpha_2(g_2 - \beta_1 g_1) \\
&= \left((1 + \eta\lambda_1)(\alpha_1 + \alpha_2\beta_1) - (1 + \eta\lambda_2)\alpha_2\beta_1\right)a_1 + (1 + \eta\lambda_2)\alpha_2 g_2 \\
&= (1 + \eta\lambda_1)\left(\alpha_1 - \left(1 - \frac{(1 + \eta\lambda_2)}{(1 + \eta\lambda_1)}\right)\alpha_2\beta_1\right)g_1 + (1 + \eta\lambda_2)\alpha_2 g_2
\end{aligned} \tag{42}$$

In the fourth step, we switched from conic coordinates $\alpha_i$ to the eigencoordinates $\beta_i$, the description in the eigenbasis of $R_B$. The last line shows the conic coordinates of the new iterate $x_1$. We choose

a learning rate $\eta < -1/\lambda_2$. Then the second coordinate is positive. The first coordinate is also positive since $(1 + \eta\lambda_2) < (1 + \eta\lambda_1)$ and $\beta_1 > 0$ because $\beta_i$, the coordinates of $g_2$ in the eigenbasis, are themselves conic coordinates of a cone spanned by the eigenvectors that $g_2$ is part of. Altogether, we conclude that the sequence $x_t$ stays within $B$. Therefore, the dynamics of the iteration scheme is entirely described by the linear map $R_B$. We estimate for a starting point $x_0$ within $B$:

$$c = \lim_{t \to \infty} \sqrt[t]{\frac{\|x_t\|}{\|x_0\|}} \leq \lim_{t \to \infty} \sqrt[t]{\|(1 + \eta R_B)^t\|} = \max_{i=1,2} |1 + \eta\lambda_i| = 1 + \eta\lambda_1 \tag{43}$$

Hence, for $\eta < -1/\lambda_2$ as chosen above, we have $c < 1$ and the sequence converges to 0 inside of $B$. Repeating this for any $B$ yields a maximal learning rate $\eta_{\max,IC}$ for inner convergence within all subregions where this is possible.

**Case 2: $v_1 \notin \partial B$ , transition to an adjacent subregion or convergence**

While we could repeat a similar calculation in conic coordinates as before, we present an alternative briefer argument here: In case the eigenvalues and $v_1$ are real, consider $f(t) = (1 - \eta R_B)^t x_0$ for $t \in \mathbb{R}$ This is a continuous map with $f(0) = x_0$ and, by power iteration, $\lim_{t \to \infty} f(t)$ approaches the direction of the largest eigenvector, which is $v_1$. Therefore, by the intermediate value theorem, there exists a minimal $t_1$ such that $g(t_1)$ lies on the boundary of the cone and $g(t)$ inside of $B$ for all $t < t_1$. Choose $t = \lceil t_1 \rceil$ (smallest integer larger than or equal to $t_1$). Then $x_t$ is outside of $B$ but, by our choice of learning rate, inside an adjacent subregion.

In case the eigenvalues and $v_1$ have an imaginary part, we can switch the basis defined by the real and imaginary part of $v_1$. There $R_B$ takes the form:

$$\begin{pmatrix} \Re(\lambda_1) & \Im(\lambda_1) \\ -\Im(\lambda_1) & \Re(\lambda_1) \end{pmatrix} = |\lambda_1| \begin{pmatrix} \Re(\lambda_1)/|\lambda_1| & \Im(\lambda_1)/|\lambda_1| \\ -\Im(\lambda_1)/|\lambda_1| & \Re(\lambda_1)/|\lambda_1| \end{pmatrix} \tag{44}$$

The last matrix is a rotation matrix. So applying this linear map $n$-times to a vector means rescaling it by a factor $|\lambda_1|^n$ and then rotating it by the angle $n \arccos(\Re(\lambda_1)/|\lambda_1|)$. Therefore, it is clear that any for any $x_0$ in the cone $B$, the cone will be left in a finite number of steps.

For completeness, there is also the possibility that $x_0$ is proportional to the second eigenvector $v_2$. By the same argument as above in Case 1, the second eigenvalue would then be negative and we would observe convergence within the learning rate bound and rate as determined in Case 1. Nevertheless, this is an edge case and does not compromise the conclusion that for any subregion we observe either convergence or transition to an adjacent subregion.

**Dynamics between Subregions** With the dynamics of single subregions classified, we can begin to glue them back together. Let $d_1$ be a vector for the first component of $Dd_1$ is zero, and $d_2$ be a vector for the second component of $Dd_2$ is zero. Then the cones spanned by $\{d_1, d_2\}$, $\{-d_1, d_2\}$, $\{d_1, -d_2\}$, $\{-d_1, -d_2\}$ along with the lower-dimensional cones in between decompose $\mathbb{R}^2$. We will denote these cones as $C$-cones. All of the cones of subregions $B$ are exactly part of one $C$ and several $B$-cones may form one $C$-cone. The statement is now within one $C$ cone the iteration scheme will either converge to 0 or leave $C$ in a finite number of steps. This is basically the same sort of statement we received for the $B$ regions. This implies that the iteration scheme cannot diverge within $C$ by jumping back and forth between subregions $B$.

To prove this, assume we have a sequence $\{x_t\}$ from our iteration scheme inside a $C$-cone that does neither leave $C$ nor converges to 0. Since $\{x_t\}$ does not converge within a subregion, it will leave any subregions it ever visits. Since the number of subregions $B$ is finite, $\{x_t\}$ must therefore revisit at least one subregion it has already been in. In two dimensions, this implies there are two adjacent subregions $B_1$ and $B_2$, and $a, b \in \mathbb{N}$ with $a < b$ such that $x_t \in B_1$ for $t = a$ and $t = b$ and $x_t \in B_2$ for $a < t < b + 1$. By construction and by assumption of no convergence, neither $B_1$ nor $B_2$ have an eigenvector inside or on their boundary.

The important observation is that inside a $D$ region there are no sign flips of $Dx$. That means even though $H$ may be described by different linear functions in the different subregions, they are glued together in a way that $H$ is continuous on $D$. The updates moving out of $B_1$ into $B_2$ is $Hx_a$ and the one moving from $B_2$ back into $B_1$ is $Hx_b$. Working in polar coordinates and recalling that the subregions are adjacent cones, this means that the angular coordinates of these two updates

must have different signs. By continuity of $H$ and the intermediate value theorem, this implies that there is an $x$ in between $x_a$ and $x_b$, in one of the subregions or their boundary, where the angular coordinate of $Hx$ is 0. Therefore, $Hx$ has only a radial component, implying $x$ is an eigenvector of the corresponding linear map and the sequence $\{x_t\}$ will converge in contradiction to the assumption that $\{x_t\}$ does not converge. As a consequence, the iteration scheme does either converge inside $D$ regions or leaves them in a finite number of steps.

A further consequence of this argument here the transition from one subregion to the next happens in one direction only, clockwise or counter-clockwise. This can be seen by repeating the above argument, where we had two different linear maps when we assumed the existence of two points with different update direction. The two linear map were continuously connected. In contrast, here we would have only one linear map for the two points, and this map is trivially continuous everywhere between, allowing us to repeat argument.

**Dynamics between $D$-Cones** To understand the dynamics between the $D$-cones, we first characterize them geometrically. Let $f_1$ be the largest eigenvector of $D$. Then we choose the vectors $d_1$ and $d_2$, which were defined by the $i$-th component of $Dd_i$ being 0 in the last subsection, to have $\langle f_1, d_i \rangle > 0$. This can be achieved by simply replacing $d_i$ by $-d_i$. Then all $d_1$ and $d_2$ lie within the same quadrant and $f_1$ lies inside the cone generated by the $d_i$. This is a direct consequence from the fact that positive definite matrices define ellipses. As our $D$ by assumption has an off-diagonal part unequal 0, the $d_i$ will not be the coordinate axes. A further consequence is that $\text{sign}(d_1) = \text{sign}(d_2) = \text{sign}(f)$.

Next, we conclude that $H(x)$ for all $x$ in $D$-cone spanned by $d_1$ and $d_2$ points to 0 in terms of their sign:
$$\text{sign}(H(x)) = -\text{sign}(Dx) = -\text{sign}(Df_1) = -\text{sign}(f_1) = -\text{sign}(x) \quad (45)$$
Similarly, we have for $x$ in the $D$-cone spanned by $-d_1$ and $-d_2$:
$$\text{sign}(H(x)) = -\text{sign}(Dx) = \text{sign}(Df_1) = \text{sign}(f_1) = -\text{sign}(x) \quad (46)$$
For the remaining cones, the last equality does not hold. Let $f_2$ be the other eigenvector of $D$ and be oriented such that it is part of the $D$ cone spanned by $d_1, -d_2$. Then for all $x$ in that cone:
$$\text{sign}(H(x)) = -\text{sign}(Dx) = -\text{sign}(Df_2) = -\text{sign}(f_2) \quad (47)$$
For all $x$ in the remaining cone spanned by $-d_1, d_2$:
$$\text{sign}(H(x)) = -\text{sign}(Dx) = \text{sign}(Df_2) = \text{sign}(f_2) \quad (48)$$
We already showed that the transitions from one subregion to the next go only in one direction. With these geometric thoughts, we can eliminate the possibility of our iteration scheme circling around forever between the $D$-cones. What is possible is going back and forth between two subregions that meet at the $d_1$ or the $d_2$ line. This can be shown by the existence of two subregion where the transitions happen only clockwise in one and counterclockwise in the other.

To show this assume without loss of generality that $d_1$ lies before $d_2$ when moving clockwise and that both lie in the first quadrant, i.e. $\text{sign}(d_1) = (+1, +1)$ Then the first subregion is given by the subregion $B_1$ whose boundary is $d_1$ and not in the cone spanned by $d_1$ and $d_2$. For $x \in B_1$ and in the first quadrant, we are in the cone spanned by $d_1, -d_2$, therefore the sign of the update direction is $(-1, +1)$. Hence the iteration scheme will cross $d_1$ clockwise. Repeating the argument for the subregion $B_2$ whose boundary is $d_2$ and not in the cone spanned by $d_1$ and $d_2$, we find that there the iteration scheme will cross $d_2$ in counter-clockwise direction. This proofs the existence of two subregions adjacent to one the $d$-lines between which the iteration scheme moves back and forth.

To describe this dynamics near the $d$-line, where the iteration scheme moves back and forth between two subregions $B_1$ and $B_2$, we work in the basis $d, e$ with $e$ being orthogonal to $d$. We denote the corresponding coordinates by $\gamma_d, \gamma_e$, and without loss of generality assume that for $\gamma_e > 0$ we move into $B_1$ First we analyze this zigzag behavior for a simplified map $H^*$ defined by $H^*(x) = B_1 x$ if $\langle \gamma_e, x \rangle > 0$ and $H^*(x) = B_2 x$ if $\langle \gamma_e, x \rangle < 0$. A drawing of this situation can be found in A.3, for the case where one of the sides moves actually away from 0. It is part of the argumentation that in this case the movement toward 0 on the other side outweighs the divergent part. Denoting $r_1 = ||B_1 d||$ and $r_2 = ||B_2 d||$, one iteration step in $B_1$ changes the coordinates as follows:
$$\begin{aligned} \Delta\gamma_d &= -\eta r_1 \gamma_d \cos(\alpha) \\ \Delta\gamma_e &= -\eta r_1 \gamma_d \sin(\alpha) \end{aligned} \quad (49)$$

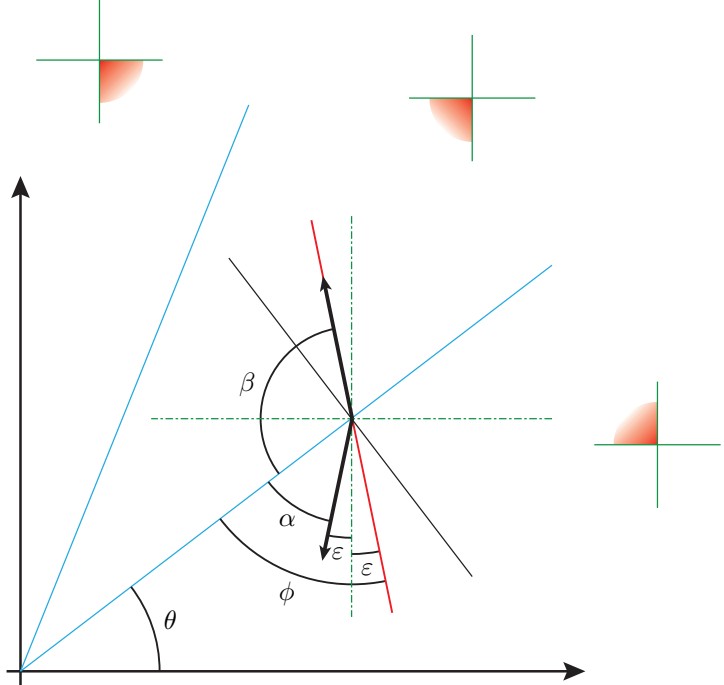

Figure 6: Geometric model for zigzag convergence. Blue lines show where the sign of the gradient field switches. The green crosses with the red-colored quadrant indicate the allowed signs of the update vector in the three regions defined by the blue lines. The red line corresponds to the flow line of the non-gradient field.

Similarly, one iteration step in $B_2$ changes the coordinates as follows:

$$\begin{aligned}
\Delta\gamma_d &= -\eta r_2 \gamma_d \cos(\beta) \\
\Delta\gamma_e &= \eta r_2 \gamma_d \sin(\beta)
\end{aligned} \tag{50}$$

For the iteration scheme to move between $B_1$ and $B_2$ endlessly, we require $\Delta_e$, the number of $B_1$ steps has to be $\kappa$ the number of $B_2$ steps, where $\kappa$ can be computed from:

$$\kappa\eta r \sin(\beta)\gamma_d = \eta r \sin(\alpha)\gamma_d \quad \text{or} \quad \kappa = \frac{\sin(\beta)}{\sin(\alpha)} \tag{51}$$

Here, we set $r = r_1 = r_2$ since along the boundary, the vector only flips a sign in one component and therefore, the lengths are the same. Consequently, the average $\Delta\gamma_d$ progress per step is:

$$\Delta\gamma_d = -\eta\Big(\frac{\kappa r}{\kappa+1}\cos(\alpha) + \frac{r}{\kappa+1}\cos(\beta)\Big)\gamma_d = -\eta F(\alpha,\beta)\gamma_d \tag{52}$$

As long as $\alpha + \beta < \pi$, the expression in the brackets $F$ gives a positive number, leading to convergence. $\epsilon$ is the angle between the $H^*(x)$ updates and the coordinate axis along which the sign flips. $\epsilon$ fulfills:

$$\alpha + \beta + 2\epsilon = \pi \tag{53}$$

By assumption of the theorem, they are not parallel, giving $\epsilon > 0$ and $\alpha + \beta < \pi$. We also introduce the angle $\phi$ between $d$ and the first coordinate axis.

$$\theta + \alpha + \epsilon = \pi/2 \tag{54}$$

To apply these ideas now to the actual vector field $H$, we note that the maximal step away from the boundary can be no more than $w_1 = (1 + \eta R_1)d$ and $w_2 = (1 + \eta R_2)d$. To receive a worst-case

estimate for the convergence rate, we assume the updates always happen with the worst possible angle $\alpha_m$ and $\beta_m$, which would lead to more movement along $e$ and less along $d$. As by continuity for $\eta = 0$, they approach $\alpha$ and $\beta$, we choose a learning rate smaller than a $\eta_{\max, ZZ}$ that is chosen such that the angle between the worst case value of $\alpha'$ and $\beta'$ on both sides of $d$ with the coordinate axis of the sign flip is still $\epsilon/2$. Then Equations 53 and 54 change to:

$$\begin{aligned} \alpha' + \beta' + \epsilon &= \pi \\ \theta + \alpha' + \epsilon/2 &= \pi/2 \end{aligned} \tag{55}$$

As still $\alpha' + \beta' < \pi$, our iteration scheme is still convergent through 52. The $\alpha'$ and $\beta'$ can be written as a function of $\phi$ and $\theta$:

$$\begin{aligned} \alpha' &= \pi/2 - \epsilon/2 - \theta \\ \beta' &= \pi/2 - \epsilon/2 + \theta \end{aligned} \tag{56}$$

With that we can give an upper bound on the convergence rate, using that $x$ is bounded by $\gamma_d$.

$$c = \lim_{t \to \infty} \sqrt[t]{\frac{\|x_t\|}{\|x_0\|}} \leq 1 - \eta F(\alpha', \beta') = 1 - \eta F(\alpha'(\theta, \epsilon), \beta'(\theta, \epsilon)) \tag{57}$$

By again using the geometry of ellipses, we can further estimate $\theta$: As mentioned $d$ lies between the eigenvector $f_1$ of $D$ and the coordinate axis. As the ellipse defined by $D$ becomes more elongated by increasing the condition number $K$ of $D$, $\theta$ increases with $K$. Using this information together with the functional form of $F$, we can then give the following formula for $c$, where $G > 0$ through $K > 1$ and $\epsilon > 0$, and $G$ fulfills $G(f_1, K_1, \epsilon) > G(f_1, K_2, \epsilon)$ if $K_1 > K_2$ as this leads to a better $\theta$.

$$c \leq 1 - \eta G(f_1, K, \epsilon) \tag{58}$$

**Final Estimates** Putting it all together, we choose $\eta_{\max}$ as the minimum of $\eta_{\max, \text{AT}}$, $\eta_{\max, \text{IC}}$, $\eta_{\max, ZZ}$. Then for $0 < \eta < \eta_{\max}$, the construction in our proof is valid and after a finite number of steps, we either reach one of the two asymptotic situations of convergence. The convergence rate is bounded by the maximum of $c \leq 1 - \eta G(f_1, K, \epsilon)$ for zigzag convergence and $c \leq 1 + \eta \lambda_1$ for convergence within one subregion.

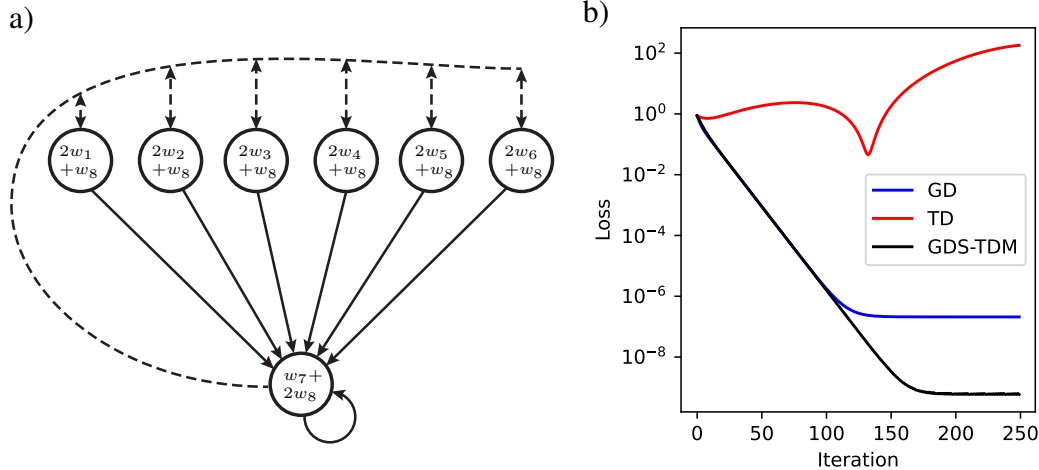

Figure 7: a) Schematic representation of Baird's counterexample with 7 states. b) Loss as a function of iterations.

## B    BAIRD'S COUNTEREXAMPLE

As another test case, we present the version of Baird's counterexample provided by Sutton & Barto (2018), which is likely the most famous example associated with TD's divergence. Figure 7a shows a schematic drawing of this Markov zero reward process along with the structure of the used function approximation. The transition probabilities of the policy to be estimated are 1 on the solid arrows and 0 elsewhere. The transition probabilities of the behavior policy used for off-policy training are $1/7$ on the solid arrows, $6/7$ on the dashed arrows, and 0 elsewhere. We use a discount factor of 0.99 and train on full batches to avoid stochastic effects of mini-batch training.

Figure 7b shows the loss (Bellmann error) over training iterations for GD, TD and GDS-TDM. At the beginning, GD performs decently before then stagnating around $10^{-6}$. This is due to ill-conditioning as an explicit calculation of the condition number of the Hessian reveals; its value is $1.3 \cdot 10^4$. TD diverges, which is due to the off-policy training procedure. GDS-TDM behaves similarly to GD in the beginning, decreasing the loss. When GD reaches is plateau, GDS-TDM continuous to decrease the loss. Later GDS-TDM stagnates as well but at a much better value of roughly $10^{-10}$. It is worth mentioning that at some point we expect the loss not to decrease any further due to floating point precision and the presence of non-zero solutions. The latter exists in any underdetermined linear system; here, we use 8 variables to learn 7 values.

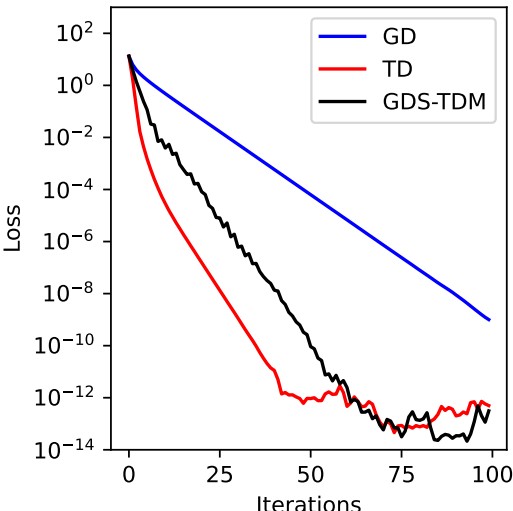

Figure 8: Loss over iterations for 3x3 Gridworld.

## C  3X3 GRID WORLD

We present an additional variation of our Grid World experiment where we reduce the size from 10x10 to 3x3. For that case, TD converges and we have a reference of what the convergence speed of TD actually is. Consequentially, we can test whether our method GDS-TDM can indeed reach the same speed as TD in higher dimensions as well. The results are shown in Figure 8. We observe GD is slower than TD, as expected; however, GD is not impractically slow since the condition number for this low-dimensional Grid World is still small. GDS-TDM lies between GD and TD. After the burn-in phase, GDS-TDM achieves a similar asymptotic speed as TD. This is an encouraging result as the theoretical analysis in our work was also focused on the asymptotic convergence rate.

