# OpenReview forum: "Temporal Difference Learning: Why It Can Be Fast and How It Will Be Faster"
_ICLR.cc/2025/Conference — ICLR 2025 Poster_

### Official Review · Reviewer_XMHR · 2024-10-27

**Soundness:** 3
**Presentation:** 4
**Contribution:** 3
**Rating:** 8
**Confidence:** 3

**Summary:**

The paper deals with convergence problems in reinforcement learning. The convergence problems of temporal difference (TD) learning and the sometimes very slow learning behavior of gradient descent (GD) are considered. The paper introduces a new way of looking at the problems, which is based on the consideration of the size of eigenvalues. An explanation is provided and a simple modification is presented which is a synthesis of TD and GD. For this purpose, the sign of the GD update vector and the magnitude of the TD update vector are used. The considerations are supported by proofs and illustrated application to small but pathological examples.

**Strengths:**

* The paper is excellently written. A real pleasure to read.
* If everything the authors present is correct, then this is a groundbreaking work that solves a problem that has been unsolved since Baird's time.

**Weaknesses:**

I cannot identify any weaknesses.

I must say, however, that I am not capable of reliably verifying that all the claims are correct.

Minor suggestions:
* “where where typically” -> “where typically”
* “equation (2)” -> “Equation (2)”
* “section 3.1” -> “Section 3.1”
* Would not be “representation of Baird’s counterexample” more natural, than “representation of the Baird’s counterexample”?

**Questions:**

* Why is the advantage of GDS-TDM not demonstrated using difficult, high-dimensional benchmarks?
* What should be considered when using GDS-TDM together with neural networks as Q-function in difficult, high-dimensional environments?
* Are the ideas from [1] relevant for the present work?
* Should [1] be cited and discussed?

[1] Wang & Ueda, Convergent and Efficient Deep Q Network Algorithm, 2022

---

> ### Author Response · Authors · 2024-11-21
> **Reponse to Reviewer XMHR**
>
> Dear Reviewer,
>
> Thank you for kind words and your questions. We will correct the mentioned typos.
>
> To answer your question, with our choice of experiments, we picked these specific environments to have a close consistency check on our theoretical framework, which is the main contribution of work. Results on higher-dimensional nonlinear networks would not allow any conclusions to be drawn on the correctness of our theory regardless of the actual result. Nevertheless, it would be an interesting direction that we hope to pursue in the future.
>
> It is important to mention again that we treat a topic with deep inconsistencies between theory and experiments: Theoretically TD can diverge, experimentally it still often finds better solutions than provably convergent methods. And despite a large body of empirical research, this behavior is still unexplained. From a statistical perspective, causal relations are established through a strong theoretical framework and controlled experimentation, and this is what we aim to provide. However, as an additional experiment, we will add Baird's original counterexample, where our method provides encouraging results.
>
> When intending to apply our methods to established RL methods, one would have to identify the value estimation part in the algorithm. Such a component exists in most RL algorithms; this is why our work is of fundamental importance. One then would need to replace it by our update rule. As our theoretical description was focused on linear approximations, one should be aware that additional adjustments might be necessary for nonlinear function approximation. As said, we hope to address such questions as future work.
>
> Regarding your reference [1], we will add this our related work as we aim to represent the diverse attempts to treat the divergence issue of TD and appreciate any hints on related work that might be relevant. They analyze a different approach that is based on introducing and updating a second set of parameters that then leads to updates on the actual network parameters. Their analysis appears not specific to particular ways of updating parameters, so the underlying updates can be GD, TD, or others. Therefore, we consider this work orthogonal to the methodological part of our work. We believe the most valuable part in our work is the theoretical framework of how convergence speed of TD methods can be investigated. Their paper and many others could use our work as a blueprint to provide theoretical arguments about whether their proposed methods will have improved convergence speed.
>
> We appreciate your review and are more than happy to engage in further discussions should you have any follow-up questions.

---

> > ### Comment · Reviewer_XMHR · 2024-11-24
> > **References**
> >
> > In the references, the arXiv source should be replaced by the corresponding conference or journal publication where possible. For example, “Convergent and Efficient Deep Q Learning Algorithm” was published at ICLR2022.

---

> > > ### Author Response · Authors · 2024-11-25
> > >
> > > Dear Reviewer,
> > > Thank you again for your feedback and for pointing out the issues with the references. We have corrected the references in the updated version.

---

> ### Comment · Area_Chair_EfJY · 2024-11-23
> **From AC,**
>
> Reviewer XMHR: if possible, can you reply to the rebuttal?

---

### Official Review · Reviewer_m5YS · 2024-10-29

**Soundness:** 3
**Presentation:** 3
**Contribution:** 3
**Rating:** 6
**Confidence:** 3

**Summary:**

This paper does a comparative analysis of the parameter updates when using gradient descent (GD) and temporal difference (TD) approaches. They focus on the one-step Bellman equation as the optimization target in both cases. In this analysis, the paper posits that GD methods have a high condition number, leading to slow convergence. It then focuses on TD updates when the Hessian matrix involved in the update is a combination of symmetric and skew-symmetric matrices.

Here the paper shows that as the skew-symmetric component of the Hessian grows, the eigenvalues start converging. With the eigenvalues closer together, the paper shows that the iteration scheme will progress more uniformly and lead to faster convergence. But this speed-up is capped by the largest learning rate that can be used.

The above analysis leads to a new method that uses the sign of GD and the magnitude of TD, called GDS-TDM in this paper. Theoretical analysis here shows that there exists a maximal learning rate under which convergence in guaranteed, and also specified the rate of convergence.

Finally, empirical evaluation on a simplified version of Baird's counter-example and a 10x10 grid world show stable learning, faster convergence, and lower error when using GDS-TDM compared to GD or TD

**Strengths:**

* The paper states that they have two contributions: theoretical analysis validating why TD can converge fast, and a unification of GD and TD that seeks to combine the positive attributes of both. From my perspective, both of these contributions are partially justified. I go into more detail on the positives below and where I might need more convincing in the following sections.
* This paper presents an interesting analysis of the GD and TD updates based on their condition number and convergence rates. The breakdown of the TD Hessian into symmetric and skew-symmetric matrices seems to be novel and leads to the subsequent analysis in the paper.
* Figure 1 is well done and gets the idea across well.
* Section 3, which focuses on the first contribution, does a good job in analyzing a simpler problem and then generalizing the result in Theorem 1. The analysis of the eigenvalues as the Hessian is skewed more in Theorem 2 is interesting and communicated well via Figure 2.
* I have checked the proofs of Theorems 1 and 2 and I did not catch any major issues (minor quibbles in the Questions section).
* The proposed modification in Section 4 seems to be simple, and the experiments bear out some of the qualities that the paper advertises, namely, faster convergence than GD and more stable learning than TD.
* The proposed modification is extremely simple.

**Weaknesses:**

Overall I really like the idea behind the paper, but I feel like there are bits and pieces that left me feeling confused. Below I try to write down these bits:
* As mentioned briefly above, I am not completely convinced that the contributions have been sufficiently justified. I appreciate that while this is a theoretical paper, the authors have attempted to empirically validate their technique. But the experiments both seem to show situations where TD is unstable, and GD is slow, and how GDS-TDM converges more stably compared to TD, and faster than GD. What of a setting where TD converges to the right solution? How does GDS-TDM compare there? Also, it is unclear why TD diverges in the grid-world example. The data is on-policy (from the optimal policy while performing policy evaluation), with what seems like not undue state aliasing. Why then does TD diverge?
* In the sub-section *Ill-Conditioning in Reinforcement Learning*, the paper indicates why GD methods have a large condition number with delayed rewards. Though it only refers to a reference book and doesn't really explain the mechanism by which the condition number is so large. Could you provide a brief explanation as to why the condition numbers would be large with delayed rewards? It would help a reader to better grasp the problem.
* The thread leading to Equation 4 is abruptly cut off with no clear takeaway for a non-expert reader. What does the update shown in Equation 4 mean for the condition number, or for the optimization problem? Why does it fail?
* I wasn't able to follow some of the notation in Section 4. I have added particular confusions in the Questions section.
* There are some other notation typos and confusions. Clarifying in the Questions section.
* End of section 1, Typo: `demonstrate with a simple method *what* a unification of GD and TD preserving their positive attributes can look like`.
* The related work has captured a lot of important references, particularly focusing on the unification of GD and TD, but has missed certain recent approaches that attempt to analyze or mitigate the instability of TD by empirically regularizing the gradient [2], or reframing the problem via an optimization lens [1]. These references might be useful to look at. I am not suggesting they be added to the related work, since it is clearly organized in a way that does not seek to exhaustively reference all previous attempts to improve the TD update.
[1] Asadi, K., Sabach, S., Liu, Y., Gottesman, O. and Fakoor, R., 2024. Td convergence: An optimization perspective. Advances in Neural Information Processing Systems, 36.
[2] Kumar, A., Agarwal, R., Ma, T., Courville, A., Tucker, G. and Levine, S., DR3: Value-Based Deep Reinforcement Learning Requires Explicit Regularization. In International Conference on Learning Representations.

**Questions:**

* Beginning of Section 3, The Hessian for TD is correctly identified as not being symmetric. Why is the conclusion that the subsequent matrix has a skew-symmetric component, and not any other type of asymmetry? Perhaps a quick clarification here might be helpful.
* In equation 8, where does the $\eta$ come from? Is it the learning rate? I thought we were looking for the eigenvalues of $H$, and not $1 - \eta H$. Which brings me to the possible incorrectness in the value of $\lambda_{1/2}$. Where does the $1$ come from?
* Equation 9 seems to have a misplaced $k$. Is it supposed to be $t$?
* Does $I_k$ in theorem 1 indicate an identity matrix?
* definition of $S$ in Theorem 3 confuses me a little bit. Does it imply [-1, -1], [0, 0], [1, 1] are all valid values for s?
* This theorem is for 2 dimensional matrices and hence only two parameters. What would be needed for a more general proof?
* How many times was the experiment repeated, what is the 95% confidence interval in Figures 4 (c) and 5 bottom?
* The proposed solution is really simple. Could you elaborate more on why these specific experiments were chosen? Perhaps experiments in more canonical RL tasks like mountain car or cart pole which have implementations online could be easily evaluated with the new technique? This would add a comparison in a control setting where TD does not diverge, where the discount factor is not 1, where the function approximation is established, and compare the empirical speed of learning and final solution reached for both TD, GD, and GDS-TDM.

---

> ### Author Response · Authors · 2024-11-21
> **Reponse to Reviewer m5YS - Part 1**
>
> Dear Reviewer,
>
> Thank you for reviewing our paper and sharing your thoughts. Below, you can find our answers to the questions you asked in the weaknesses (W) and questions (Q) sections. We will fix the typos you mentioned and omit them in the list below.
>
> W1: Question on our choice of experiments: What of a setting where TD converges to the right solution? How does GDS-TDM compare there? Also, it is unclear why TD diverges in the grid-world example. The data is on-policy from the optimal policy, why then does TD diverge?
>
> Thank you for raising this crucial point, which we will further clarify in our paper. Most importantly, TD does not provably converge, and hence test cases should distinguish "convergent TD" and "divergent TD" cases. Our paper argues theoretically that convergent TD offers a speed advantage and we present an idea of how to preserve the mechanism behind this speed advantage for divergent TD with a new method. Therefore, in a scenario where TD is already convergent, we would not expect our method to perform better as there is "nothing left to fix". Rather, our method was developed for cases where TD diverges, and hence we primarily choose to investigate its behavior for experiments that show diverging behavior. In the grid world example, we predict the values of the optimal policy. However, the state pairs are not samples from this policy, i.e. this is an off-policy case. Therefore, TD divergence is a real possibility and does not contradict the theory. In more detail, the state pairs in our setup are all equally likely whereas for an optimal policy, the state pairs closer to the terminal states are encountered more often. For brevity, we didn't mention the on- and off-policy topics until the related work section, but your question highlights the importance of this distinction, and we will clarify it in the upcoming revision. As your question Q8 also concerns the experiments, we will further comment on our choice of experiments there.
>
>
> W2: In the sub-section Ill-Conditioning in Reinforcement Learning, the paper indicates why GD methods have a large condition number with delayed rewards. Though it only refers to a reference book and doesn't really explain the mechanism by which the condition number is so large. Could you provide a brief explanation as to why the condition numbers would be large with delayed rewards?
>
> While not present in the mentioned subsection, we cite one work that explicitly studied connections between condition numbers and Reinforcement Learning (RL) parameters in the related work section. When we discuss the role of ill-conditioning in RL in our background section, we tried to provide some intuition on that issue while remaining concise. The linear system of the Markov reward process we defined in this subsection is one of the most studied linear problems, commonly known under the name Poisson problem. The reference only provides the scaling of the condition number of this particular linear system but adds nothing in terms of RL or delayed rewards.  We will clarify this in our updated version of the paper.
>
> One way to understand this behavior is that delayed rewards that connect states over many time steps lead to larger coupled linear systems and it is well-established in linear algebra that larger matrices typically have larger condition numbers. Another way would be to compare our Markov reward process against one where all n-state transition immediately connect to the terminal state. That would definitely be free of delayed rewards and the matrix in the Bellmann equation would be the identity matrix, i.e. perfectly conditioned. It is important to keep in mind that delayed rewards are only one way how condition numbers can increase.

---

> ### Author Response · Authors · 2024-11-21
> **Reponse to Reviewer m5YS - Part 2**
>
> W3: The thread leading to Equation 4 is abruptly cut off with no clear takeaway for a non-expert reader. What does the update shown in Equation 4 mean for the condition number, or for the optimization problem? Why does it fail?
>
> Equation 4 is a gradient update.  As such, its success or failure depends on the condition number, as we explained in the subsection 'Optimization theory'. The form of this update is a product of the TD error and the function approximation of values, as dictated by the chain rule of derivatives. Therefore, the overall conditioning can be expected to be ill since the TD error is already ill-conditioned, as we explained in the subsection 'Ill-Conditioning in RL'.  On top, function approximation isn't usually well-conditioned either.  We should mention that the condition number of a matrix product does generally not need to be higher than the matrix factors it is made of.  E.g. the condition number is one when the two matrix factors are a matrix and its inverse, or unchanged if one of the two factors is an identity matrix. But in the practical relevant scenario of two matrix factors independent of each other, the overall conditioning will with high probability be worse. In our revision, we will add this additional explanation and make the relation to the previous subsections clearer.
>
> W7: The related work has captured a lot of important references, particularly focusing on the unification of GD and TD, but has missed certain recent approaches that attempt to analyze or mitigate the instability of TD by empirically regularizing the gradient [2], or reframing the problem via an optimization lens [1].
>
> We are eager to keep up with recent developments in RL and also offered some references of non-gradient updates beyond the RL field, as we think it is important to bridge different fields where the underlying mathematical problems are similar. [1] studies generalizations of linear TD's on policy convergence, such as for non-square losses or special cases of nonlinear function approximators. [2] doesn't study the divergence issue directly but rather the properties of solutions when TD succeeds and how to find better ones. Our work differs from these papers by providing a theoretical explanation for the faster convergence of TD training. Any suggestions for works whose relation to our work should be discussed are more than welcome and we are happy to add them in our updated draft.
>
> Q1: Beginning of Section 3, The Hessian for TD is correctly identified as not being symmetric. Why is the conclusion that the subsequent matrix has a skew-symmetric component, and not any other type of asymmetry?
>
> Any matrix $A$ can be written as sum of its symmetric part $(A+A^T)/2$ and skew-symmetric part $(A-A^T)/2$. Hence a non-symmetric matrix has a non-zero skew-symmetric component in such a decomposition. When we decompose $H_{TD}$ in that way, it is actually less a conclusion than it is a preparation for our mathematical treatment of TD's convergence speed. To avoid any confusion, we should clarify that this is not a Hessian in the strict mathematical sense as there is generally no associated scalar potential for TD updates.
>
> Q2: In equation $8$, where does the $\eta$ come from? Is it the learning rate? I thought we were looking for the eigenvalues of $H$, and not $1-\eta H$. Which brings me to the possible incorrectness in the value of $\lambda_{1/2}$. Where does the $1$ come from?
>
> $\eta$ is indeed the learning rate and the eigenvalues in Equation $8$ are the eigenvalues of the iteration operator $1-\eta H$ from Equation $6$, as you correctly suspected. Since we illustrated the optimization trajectories in Figure $1$, it was also our intention to show the eigenvalues of $1-\eta H$. The observations mentioned in this section, that the eigenvalues converge and become complex at some point, hold equally for $H$ and $1-\eta H$. Nevertheless, we see that we didn't phrase this well in the paragraph before Equation $8$ and will change our manuscript accordingly. Thank you for pointing this out. We also double-checked our theorems to make sure it is clarified if we consider eigenvalues of $H$ or $1-\eta H$.
>
> Q3: Equation 9 seems to have a misplaced $k$. Is it supposed to be $t$?
>
> Yes, thanks.
>
> Q4: Does $I_k$ in theorem 1 indicate an identity matrix?
>
> No, $I_k$ is the imaginary part of the $k$th eigenvalue of $H$. This is stated in line $1$-$2$ of Theorem 1.

---

> ### Author Response · Authors · 2024-11-21
> **Reponse to Reviewer m5YS - Part 3**
>
> Q5: Definition of $S$ in Theorem 3 confuses me a little bit.  Does it imply $[-1, -1], [0, 0], [1, 1]$ are all valid values for $s$?
>
> Yes, these are indeed valid values for $s$.  The complete set is $S=\{(-1,-1),(-1,0),(-1,1),(0,-1),(0,0),(0,1),(1,-1),(1,0),(1,1) \}$.  We noticed we forgot to add $\forall i=1,2$ in the definition of $S$, which may have caused some confusion. Apart from that, the definition might seem a bit unusual at first, but the elements of $S$ simply serve to indicate the areas where the GD and TD update vectors don't change their sign.
>
> Q6: Theorem 3 is for two-dimensional matrices and hence only two parameters. What would be needed for a more general proof?
>
> Many of the complications arise from the non-differentiablity of the GDS-TDM vector field that is constructed from smooth GD and TD vector fields.  Most convergence proofs in optimization rely on such properties; therefore, such proofs couldn't serve as blueprint or even inspiration.  Our proof works roughly by subdividing the space into ever smaller regions, with the lowest layer having regions described by a linear map. There, a generalization of our characterization of the dynamics is straightforward. The challenging parts begin when be piece these smaller regions back together to characterize the global dynamics.
>
> In two dimensions, we were able to eliminate certain kinds of divergent cyclic behavior between these regions by assigning them either clockwise or counter-clockwise movement, depending on how optimization trajectories look like within a region. In higher dimensions, another approach would be needed, potentially a proof by contradiction via some fixed point theorem to eliminate the possibility of such diverging cyclic trajectories.
>
> Q7: How many times was the experiment repeated, what is the 95 percent confidence interval in Figures 4 (c) and 5 bottom?
>
> The experiments were not repeated because both are convex optimization tasks. Hence, there are no local non-global minima or other complications that depend on initialization. The initial parameter values can change the initial loss values but asymptotic properties such as convergence and convergence rates are unaffected. Therefore, the shown results should not be considered lucky outliers, but are representative for the behavior of the different methods.
>
> Q8: The proposed solution is really simple. Could you elaborate more on why these specific experiments were chosen? Perhaps experiments in more canonical RL tasks like mountain car or cart pole which have implementations online could be easily evaluated with the new technique? This would add a comparison in a control setting where TD does not diverge, where the discount factor is not 1, where the function approximation is established, and compare the empirical speed of learning and final solution reached for both TD, GD, and GDS-TDM.
>
>
> Our experiments are motivated by the fact that the topic we treat has a considerable gap between theory and experiments. As outlined in our introduction, theoretically TD can diverge, experimentally it still often finds better solutions than provably convergent methods. And despite a large body of empirical research, this behavior is still unexplained. From a statistical perspective, causal relations are established through a strong theoretical framework and controlled experimentation, and this is what we provide. The 2-state experiment is a clear consistency check on the convergence proof of our method with all stochastic elements eliminated. The grid world example gives a glimpse on the question if such a proof could also be formulated in higher dimensions. Experiments on more complex environments are no doubt interesting, no less for us as authors, but regardless the result, they would not provide any insights about the correctness of our theoretical derivation, and we would again face the same gap between theory and experiments.
>
> However, we followed the proposal of one of the other reviewers to include Baird's 7-star problem as an additional higher-dimensional experiment since it is probably the most studied example for the convergence properties of TD. It is promising to see GDS-TDM performing best in this test. Further experiments would also draw the focus further away from the first part of paper, our theoretical explanation of the speed advantage of TD. This is probably the most valuable part of our work as, to the best of our knowledge, nobody has previously provided an explanation for this advantage.
>
> Thank you again for your many suggestions and questions. In case of follow-up questions, we'll happily discuss further.

---

> > ### Comment · Reviewer_m5YS · 2024-11-23
> > **Response to Author Comments**
> >
> > I thank the authors for their detailed response. Assuming the manuscript is updated to address the points of confusion that the authors clarified in their response, most of my minor issues with the paper will be addressed.
> > I am content with not including the citations I suggested in the paper. The author response has clarified why they might not be relevant here.
> >
> > The paper is an interesting contribution to the theoretical analysis on TD convergence, and as such I believe it is worthy of publication. I would still encourage the authors to consider some more empirical evaluations. Specifically, I repeat my assertion that evaluation on an example where TD converges, to show practically that the proposed technique can match the convergence speed of TD, would be useful. Additionally, while the paper focuses on theoretical analysis, if the proposed algorithm could be shown to be practically useful (even on a fairly simple problem like grid world), it could have higher impact by garnering attention from researchers who focus on pushing the empirical frontier of reinforcement learning.

---

> > > ### Author Response · Authors · 2024-12-04
> > >
> > > Dear Reviewer,
> > >
> > > Thank you for your response.
> > >
> > > From your initial review, we were unsure why the behavior of our method in a
> > > convergent TD scenario was of interest to you, but your response has clarified this.
> > > As we understand your argument, in a convergent TD case, there is an actual
> > > reference of how large the TD speed concretely is. Therefore, such a setup would
> > > allow us to characterize the speed-up from two sides, for example ‘faster than GD’
> > > and ‘almost as fast as TD’. In contrast, our divergent TD experiments only allow us
> > > to assert that our method is 'faster than GD'. We agree with your suggestion that
> > > such an experiment could be helpful to better understand how our method relates to
> > > both GD and TD.
> > >
> > > We conducted such a convergent TD experiment on a 3x3 grid world case, as
> > > divergence issues of TD appear to be rarer on smaller grid worlds. Naturally, this
> > > setup is not as ill-conditioned as the 10x10 version in our paper, but still suited to
> > > test whether our method can approach the speed of TD. Since PDF uploads are not
> > > available at this time, we report the results in form of the following table:
> > >
> > > | Step | GD | TD | GDS-TDM
> > > | -------- | ------- | ------- | ------- |
> > > | 0 | 1.3e+01 | 1.3e+01 | 1.3e+01 |
> > > | 10 | 4.8e-01 | 2.9e-05 | 3.9e-03 |
> > > | 20 | 5.2e-02 | 1.6e-07 | 8.2e-05 |
> > > | 30 | 5.5e-03 | 1.1e-09 | 6.0e-07 |
> > > | 40 | 5.9e-04 | 1.1e-11 | 1.3e-08 |
> > > | 50 | 6.3e-05 | 9.3e-13 | 9.2e-11 |
> > > | 60 | 6.8e-06 | 4.7e-13 | 2.4e-12 |
> > >
> > > Ignoring the first (roughly 10) burn-in steps, and focusing on the region with
> > > asymptotic behavior, we observe convergence rates of 0.80 for GD, 0.61 for TD, and
> > > 0.64 for GDS-TDM. A convergence rate of 0.8 means that if the loss value is L at the
> > > current step, then in the next step it will decrease to 0.8*L. These results align with
> > > the theoretical insights in our paper: for this 'convergent TD case' GDS-TDM
> > > converges clearly faster than GD, and approaches the convergence speed of TD. We
> > > will include this experiment with the corresponding loss curves in our next revision of
> > > our paper.
> > >
> > > Thank you once again for your suggestions.

---

> ### Comment · Area_Chair_EfJY · 2024-11-23
> **From AC.**
>
> Reviewer m5YS: if possible, can you reply to the rebuttal?

---

### Official Review · Reviewer_1w4Z · 2024-11-02

**Soundness:** 3
**Presentation:** 3
**Contribution:** 3
**Rating:** 6
**Confidence:** 2

**Summary:**

The paper proposes a theoretical analysis to understand the speed benefits of TD and the convergent properties of GD. It proposes a method that combines the benefits of each to have a faster and convergent algorithm. It presents a theoretical analysis and empirical results on toy domains.

**Strengths:**

- The proposed method of combining TD and GD is simple and reasonable.
- The paper includes nice visualizations to understand the intuition behind the trajectories of the algorithms.
- The toy domains illustrate the idea reasonably well. It is nice to include results on Figure 4.

**Weaknesses:**

- While not an empirical paper, I think some comparison to methods like GTD (gradient TD) would be useful to get a sense of the rate of convergence of other similar algorithms. I think just evaluating TD and GD in Section 5 is limited.
- Perhaps it would be useful to also evaluate the proposed methods on other toy domains known to exhibit convergence like the full Baird counter example like Figure 11.1 from Sutton and Barto [1] and Figure 1a from [2].
- I think the paper can do better at improving intuition in terms of language. The attempt of 3.4 is nice of giving the complete picture, but I think the broader implications are still missing and the details are still too narrowly focused on technical details. It would be nice to include some broader implication on future design of such algorithms based on the theoretical results.

[1] http://incompleteideas.net/book/RLbook2020.pdf
[2] A Kernel Loss for Solving the Bellman Equation. Feng et al. 2020.

**Questions:**

- While there are some details in the related works, I am having a tough time placing this work in the context of others. Currently, the paper is written as though no one else has worked on this problem since Baird’s 1999 papers. Is this actually true? While the proposed method may be new, Theorem 1 and 2 seem quite fundamental and it's interesting that other works have not explicitly tried to address this. This question may be due to just my unfamiliarity with the literature, and so I would defer to other reviewers on the value of the proposed insights.
- The paper cites Gallici et al. 2024 for their use of layer norm to address TD divergence. I am curious whether the authors think this itself is sufficient for getting both a convergent and fast algorithm? If layernorm enables TD to converge, we don’t necessarily need to rely on GD’s convergence benefits right?

---

> ### Author Response · Authors · 2024-11-21
> **Reponse to Reviewer 1w4Z**
>
> Dear Reviewer,
>
> Thank you for your interest and giving feedback on our work.
>
> Your first suggestion was to include GTD as a further comparison method in our experiments. Here it is important to point out that our experiments are on deterministic environments, for which GTD simplifies to plain Gradient Descent (GD), as the next state sampling process inherent in GTD methods always leads to the same outcome. GTD are specifically designed for another difference between GD and Temporal Difference (TD) methods, namely that they can have different fixed points on stochastic environments. We focus on the convergence speed difference between GD and TD, and so deliberately chose deterministic systems to offer a clear ground without other complications. Along the same lines, we think the experiment from your reference [2] would unfortunately not be helpful due to its stochastic rewards and transitions. In contrast, we followed your advice to include Baird's counterexample. It doesn't have the issue with different fixed points and is probably the most famous example for TD's divergence. It is encouraging to see that GDS-TDM performs best on this experiment and will include the details in our upcoming revision.
>
> Another suggestion was to focus more on the broader implications of our work and less on the technical details.
> It is important to highlight that there is a considerable gap between theory and experiments around TD. As outlined in our introduction,
> theoretically TD can diverge, experimentally it still often finds better solutions than provably convergent methods, and this has been unexplained for a long time despite lots of empirical research. We think the best approach to address this gap is by presenting a theoretical description that holds under mathematical scrutiny and we would argue that the technical details are valuable in achieving this.
> Our goal with the simple method in Section 4 (GDS-TDM) was to show that this understanding can indeed already benefit future design of algorithms. This method is built on our understanding of TD's speed from Section 3. We will try to clarify this further in our manuscript and also mention again that value estimation through TD is part of many practical RL algorithms. We think the fundamental character of our work is interesting enough to warrant further research that can ultimately benefit practical RL algorithms.
>
> Questions section:
>
> Q1: How to place this work in the context of others? Are there other works on this topic since Baird's 1999 papers? Are Theorems 1 and 2 unique, or have they been addressed by other works?
>
> We believe we are indeed the first to describe the speed advantage of TD theoretically. While there are other works addressing TD's divergence, which we discuss in our related work section, they focus on different aspects. For instance, as we explained above, GTD addresses another issue. We agree that Theorems 1 and 2 are indeed quite fundamental, but we could not identify any other works, even in different fields, performing a similar analysis with non-symmetric matrices.
>
> Q2: Citation of Gallici et al. 2024 for their use of layer norm to address TD divergence. Is this enough for a convergent and fast algorithm? If yes, would we need to rely on GD's convergence benefits anymore?
>
> We cited this work to show TD's divergence issue is an active area of research with diverse attempts to solve it. Regarding the convergence speed, we think it will be hard to argue for any method to have TD's convergence speed, as to the best of our knowledge, our framework is the first to show how such an theoretical argument on the convergence speed can be made. Regarding convergence, in our opinion, it makes sense to be open to the various approaches people take to tackle this issue and we will see what future research reveals. We have no specific argument if or if not layer normalization can enforce TD's convergence. But we agree with the implication that in the presence of further convergent methods, we wouldn't need to rely as much on GD.
>
> Thank you again for your review and we are happy to discuss further questions you may have.

---

> > ### Comment · Reviewer_1w4Z · 2024-11-22
> >
> > Thanks to the authors for their response. This was helpful in defining context for me. It would be great to see the preliminary results of the Baird experiment that you ran.
> >
> > Given how the scoring is done (increments of 2), I'll keep the score the same.

---

### Author Response · Authors · 2024-11-21
**Response to All Reviewers**

Dear Reviewers,

Thank you for taking the time to review our paper and sharing your thoughts. We are glad to know that all of you were able to follow our arguments. In the individual responses, you will find our answers to your questions and concerns. We also plan to upload an updated draft as soon as we incorporated your feedback. Any further discussion is welcome and we would be happy to address any additional points.

---

### Author Response · Authors · 2024-11-23
**Paper Update**

Dear Reviewers,

We have uploaded the updated version of our paper. To address your comments, we have revised several formulations for improved clarity and added additional explanations. Furthermore, we have included Baird’s counterexample in the appendix as a further experiment. All changes have been marked in blue for your convenience.

Thank you once again for your feedback on our work.

---

### Meta-Review · Area_Chair_EfJY · 2024-12-19

**Metareview:**

The paper looks into two algorithms for learning value functions: temporal difference learning (TD) and gradient descent (GD). The authors analyse both, arguing that TD if faster while GD has better convergence properties. They propose GDS-TDM, a method that takes the sign from the gradient descent update and the magnitude from the td update, combining the benefits of both approaches.

This is a strong and novel submission that proposes a simple-to-implement algorithm for a relevant problem. It also includes good visualisations on toy domains and a novel theoretical analysis.

Because of the strengths of the submission, I recommend acceptance, while strongly encouraging the authors to improve the presentation for the camera-ready version. Specifically:
- The paper implicitly realises that TD learning doesn't minimise any function, but does not emphasise this enough. Since recognising this fact is a fundamental feature of any analysis of TD learning, it has to be much more prominent.
- On a related note, when comparing with gradient descent, it has to be made much more clearly what objective (loss function) the gradient descent is using.

**Additional Comments On Reviewer Discussion:**

Like I said in the meta-review, for me the main problem with this otherwise great paper seems to be in the presentation. Since the paper is clear for some reviewers (unlike for me), I wanted to have a discussion about this. Since no reviewers participated in the discussion, I defaulted to my initial view (reluctant rejection)

Having said that, I am still totally on the fence about this paper. I really would not mind if it got accepted.

--------------------------------------------------------

Update: I have updated the meta-review to recommend acceptance, while using the meta-review to encourage the authors to make two changes to presentation.

---

### Decision · Program_Chairs · 2025-01-22

Accept (Poster)